# Structural insights into the mechanism of activation of the TRPV1 channel by a membrane-bound tarantula toxin

Chanhyung Bae[1,2†], Claudio Anselmi[3†], Jeet Kalia[1,4], Andres Jara-Oseguera[1], Charles D Schwieters[5], Dmitriy Krepkiy[1], Chul Won Lee[6], Eun-Hee Kim[7], Jae Il Kim[2*], José D Faraldo-Gómez[3*], Kenton J Swartz[1*]

[1]Molecular Physiology and Biophysics Section, Porter Neuroscience Research Center, National Institute of Neurological Disorders and Stroke, National Institutes of Health, Bethesda, United States; [2]Department of Life Science, Gwangju Institute of Science and Technology, Gwangju, Republic of Korea; [3]Theoretical Molecular Biophysics Section, National Heart, Lung and Blood Institute, National Institutes of Health, Bethesda, United States; [4]Indian Institute of Science Education and Research, Pune, Pune, India; [5]Division of Computational Bioscience, Center for Information Technology, National Institutes of Health, Bethesda, United States; [6]Department of Chemistry, Chonnam National University, Gwanju, Republic of Korea; [7]Protein Structure Research Group, Korea Basic Science Institute, Ochang, Republic of Korea

*For correspondence: jikim@gist. ac.kr (JIK); jose.faraldo@nih.gov (JDFG); swartzk@ninds.nih.gov (KJS)

†These authors contributed equally to this work

**Abstract** Venom toxins are invaluable tools for exploring the structure and mechanisms of ion channels. Here, we solve the structure of double-knot toxin (DkTx), a tarantula toxin that activates the heat-activated TRPV1 channel. We also provide improved structures of TRPV1 with and without the toxin bound, and investigate the interactions of DkTx with the channel and membranes. We find that DkTx binds to the outer edge of the external pore of TRPV1 in a counterclockwise configuration, using a limited protein-protein interface and inserting hydrophobic residues into the bilayer. We also show that DkTx partitions naturally into membranes, with the two lobes exhibiting opposing energetics for membrane partitioning and channel activation. Finally, we find that the toxin disrupts a cluster of hydrophobic residues behind the selectivity filter that are critical for channel activation. Collectively, our findings reveal a novel mode of toxin-channel recognition that has important implications for the mechanism of thermosensation.

## Introduction

The venom of poisonous animals contains an arsenal of protein toxins that target ion channel proteins to paralyze prey and induce pain by activating nociceptive sensory neurons (*Bohlen and Julius, 2012*; *Kalia et al., 2015*). These toxins, which typically contain multiple disulfide bonds that stabilize their tertiary structure (*Norton and Pallaghy, 1998*; *Pallaghy et al., 1994*), activate or inhibit ion channel proteins that open and close in response to membrane voltage (*Cestele et al., 1998*; *Milescu et al., 2013*; *Swartz and MacKinnon, 1997a*), binding of neurotransmitters (*Celie et al., 2005*; *Chen et al., 2014*; *Dellisanti et al., 2007*; *Ulens et al., 2006*) or other sensory stimuli (*Bohlen et al., 2011*; *Bohlen et al., 2010*). Thus, these small proteins are invaluable tools for investigating ion-channel structures and operational mechanisms.

**eLife digest** Humans and other mammals sense heat using a protein called the transient receptor potential vanilloid (TRPV1) channel. This protein is found in the membranes of a particular type of nerve cell, and it forms a pore that allows certain ions to pass through the membrane. Along with sensing heat, TRPV1 can also be activated by a toxin called double-knot toxin – which is found in spider venom – and by capsaicin, the active ingredient in chilli peppers.

One way to investigate how a protein works is to study its three-dimensional (3D) structure. Here, Bae, Anselmi et al. use a technique called nuclear magnetic resonance (NMR) spectroscopy to produce a detailed model of the 3D structure of double-knot toxin. This model is then combined with 3D maps of TRPV1 from previous studies to predict where the toxin binds to TRPV1. This suggests that the toxin binds to a section of TRPV1 that is buried within the membrane. Moreover, the new models highlight a 'hydrophobic' region of the TRPV1 channel that may work as the heat sensor.

Together, Bae, Anselmi et al.'s findings reveal a new way in which a toxin can bind to a target protein in membranes. The next step is to test the idea that the hydrophobic region identified in this work is the part of TRPV1 that senses heat.

Many of these protein toxins have evolved to interact with functionally important domains of ion channels that are freely accessible to water on the external side of the cell membrane, implying that toxin-channel recognition takes place in an aqueous environment. For example, the scorpion toxin charybdotoxin binds to the extracellular side of the pore of potassium channels to physically block the flow of ions (*MacKinnon and Miller, 1988*). A recent X-ray structure of a complex between charybdotoxin and the Kv1.2-2.1 paddle chimera channel revealed that this block is effected by a lysine residue that the toxin inserts into the selectivity filter of the channel, mimicking a $K^+$ ion (*Banerjee et al., 2013*). Another example is the tarantula toxin PcTx1, which activates acid-sensing ion channels (ASICs) by clamping onto helix 5 within the large extracellular domain of the channel, and inserting an Arg finger motif into a subunit interface where protons bind (*Baconguis and Gouaux, 2012*; *Chen et al., 2006*; *Dawson et al., 2012*; *Salinas et al., 2006*). The ASIC channel was also crystalized in complex with MitTx (*Baconguis et al., 2014*), a structurally distinct two-subunit snake toxin that binds to an extended region of the extracellular domain of the channel, stabilizing it in an open conformation. The general principle of toxins binding to ion channels within aqueous environments is further illustrated by X-ray structures of a glutamate-activated cation channel in complex with the cone snail toxin con-ikot-ikot (*Chen et al., 2014*), and of the extracellular domains of Cys-loop receptors in complex with cone snail or snake toxins (*Celie et al., 2005*; *Dellisanti et al., 2007*; *Ulens et al., 2006*). These striking structures not only reveal in great detail the nature of toxin-channel interactions in solution, but also provide valuable insights into the operational mechanisms of an array of ion channels. This notwithstanding, it has been recently suggested that certain types of toxins can target domains within ion channel proteins that are embedded within the lipid membrane. For example, tarantula toxins that modify the gating of voltage-activated ion channels partition into membranes and are thought to bind to the voltage-sensing domains within the membrane environment (*Alabi et al., 2007*; *Gupta et al., 2015*; *Lee and MacKinnon, 2004*; *Mihailescu et al., 2014*; *Milescu et al., 2009*; *Milescu et al., 2007*; *Phillips et al., 2005b*). Thus far, however, no structures of toxin-channel complexes have been solved for this class of toxins, and therefore our understanding of toxin-channel interactions within membrane environments is limited.

Double-knot toxin (DkTx) is a tarantula toxin isolated from the venom of a Chinese bird spider (*Bae et al., 2012*; *Bohlen et al., 2010*) that activates the transient receptor potential vanilloid 1 (TRPV1) channel. TRPV1 is a cation channel expressed in nociceptive sensory neurons that plays important roles in the transduction of noxious stimuli as well as thermosensation (*Julius, 2013*), but the molecular mechanism of heat-dependent activation has remained elusive. From a structural perspective, DkTx is intriguing because it has an unusual bivalent architecture, being comprised of two inhibitor-cysteine-knot (ICK) motifs that have been designated as the K1 and K2 lobes. DkTx is also quite hydrophobic, requiring the use of detergents to efficiently fold in vitro (*Bae et al., 2012*),

which raises the possibility that the toxin might interact with TRPV1 in a membrane environment, similar to toxins targeting voltage-sensing domains of voltage-activated ion channels. In a recent breakthrough, near-atomic resolution structures of the TRPV1 channel were solved using single-particle cryo-electron microscopy (cryo-EM), including a closed state (apo), a capsaicin-bound state, and an open state with both DkTx and the vanilloid resiniferatoxin (RTx) bound (*Cao et al., 2013*; *Liao et al., 2013*). The structure of the complex between TRPV1 and DkTx/RTx unambiguously reveals that the DkTx lobes bind to sites at the periphery of the external pore of the channel. However, the resolution of electron density maps was insufficient to reveal the structure of DkTx in atomic detail. Therefore, the binding arrangement of the two DkTx knots is unknown, as is the nature of the interactions between toxin and channel, or whether the interaction occurs in aqueous or membrane environments. How DkTx recognition ultimately results in channel opening, and how this process might relate to the mechanism of temperature-sensing, are also open questions.

In the present study, we use nuclear magnetic resonance (NMR) spectroscopy to solve the solution structure of DkTx, and the molecular-modeling suite ROSETTA together with existing cryo-EM density maps to derive high-quality structural models of the TRPV1 channel with and without DkTx bound. We also investigate the interaction of the toxin with the TRPV1 channel and lipid membranes using fluorescence spectroscopy, electrophysiological recordings and molecular dynamics (MD) simulations. Our results demonstrate that DkTx interacts intimately with TRPV1 while inserting hydrophobic residues into the surrounding lipid membrane, and provide the first example of a complex structure between toxin and channel within a membrane environment. Comparison of the improved structures of apo and toxin-bound TRPV1 reveals novel insights into the mechanism by which DkTx induces channel opening, and provides new insight into the mechanism of thermosensation.

## Results

### NMR structures of K1 and K2 lobes of DkTx

Our first objective was to solve the structure of DkTx in solution at atomic resolution through solution NMR spectroscopy. Because previous studies have demonstrated that the two lobes (or knots) of the toxin, K1 and K2, can fold independently and can also activate the TRPV1 channel (*Bae et al., 2012*; *Bohlen et al., 2010*), we synthesized K1 and K2 individually using solid-phase peptide synthesis. The full-length toxin was also produced recombinantly, with and without $^{15}$N-labeling. All constructs were folded in vitro in the presence of detergent, and correctly folded species were purified using reversed-phase HPLC. Our strategy was to first use two-dimensional (2D) $^1$H-$^1$H nuclear Overhauser effect spectroscopy (NOESY) to study each of the lobes separately, because these measurements would yield less crowded spectra and thus facilitate the assignment of a greater number of cross-peaks, which could then be translated into nuclear Overhauser effect (NOE) inter-proton distance restraints for structure determination. We then conducted additional NMR experiments using $^{15}$N labeled DkTx, taking advantage of the separation of overlapped spectra in 2D $^1$H-$^1$H NOESY using $^{15}$N, and compared the backbone proton resonances of K1 and K2 with the full-length protein.

Complete proton resonance assignments for K1 and K2 were made using traditional 2D NMR sequential assignment techniques (*Wüthrich, 1986*) (*Figure 1—figure supplement 1*). Using the proton chemical shift values of isolated K1 and K2 as reference, proton resonances in DkTx were readily identified (*Figure 1—figure supplement 1,2*). The backbone proton chemical shift values of DkTx were found to be nearly identical to those measured for K1 and K2 separately, except for a few residues in the linker region and in the N- and C-termini of the toxin (*Figure 1—figure supplement 3*). We can therefore conclude that the structures of isolated K1 and K2 are highly similar to those in full-length DkTx.

The structures of K1 and K2 were determined using the inter-proton distance restraints deduced from the NOESY data (*Figure 1—figure supplement 1*), in addition to backbone dihedral angle phi restraints estimated from DQF-COSY spectra (*Kim and Prestegard, 1989*), and disulfide-bond restraints inferred from homology with other ICK toxins of known structure. Using the simulated-annealing method and energy function in Xplor-NIH 2.37 (*Schwieters et al., 2006*; *Schwieters et al., 2003*), we derived a set of 20 energy-minimized structures for each knot that fulfill the NMR restraints, as well as generic protein-geometry and other knowledge-based restraints (*Bermejo et al., 2012*). Statistical analysis of these ensembles (*Figure 1—figure supplement 4*)

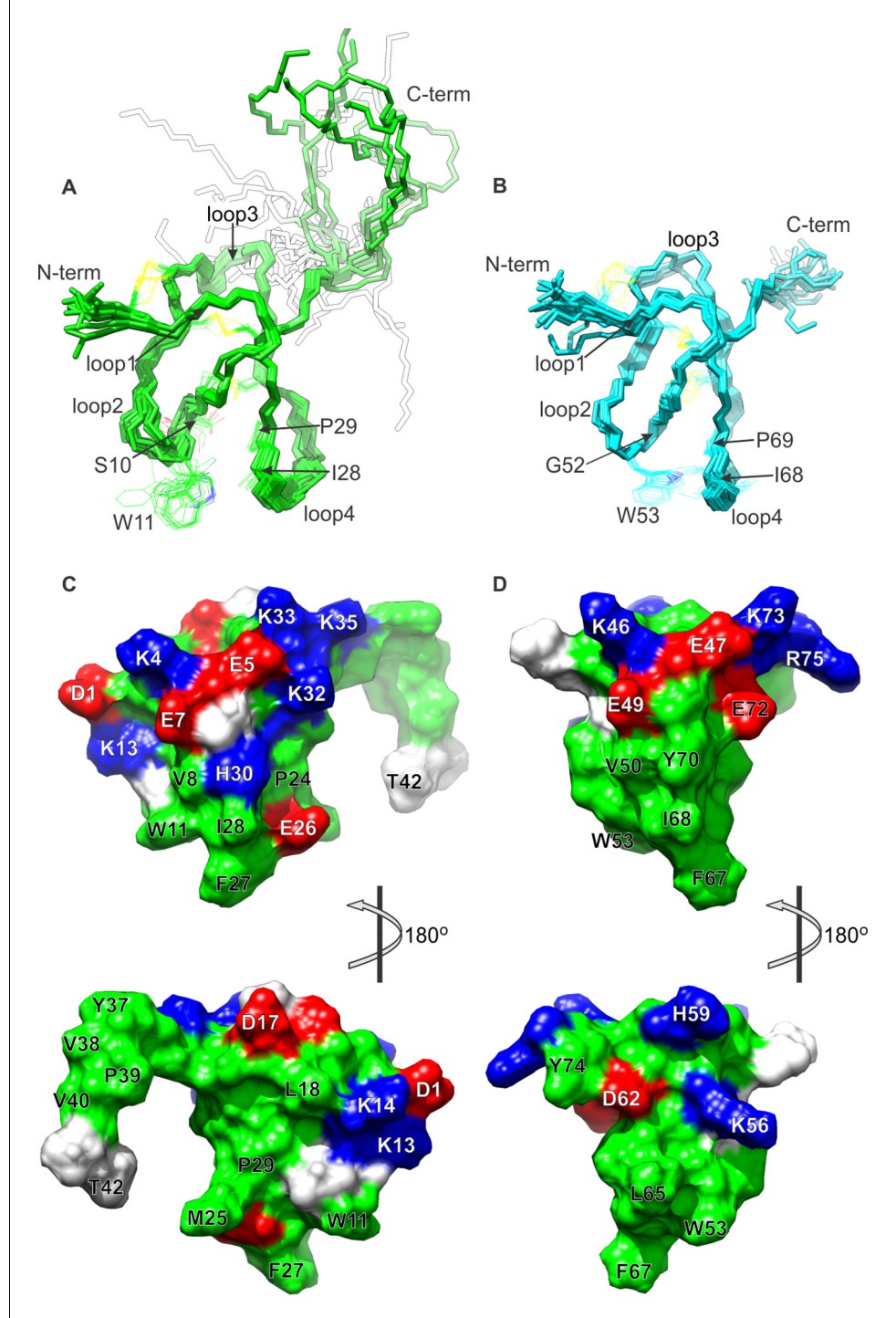

**Figure 1.** NMR solution structures of K1 and K2. (**A,B**) Ensemble of backbone structures of K1 (**A**), in green, and K2 (**B**), in cyan. The 20 lowest-energy structures out of 100 energy-minimized structures are shown (PDB entry 2N9Z for K1 and 2NAJ for K2). The distinct orientations of W11 and W53, as well as their interactions with residues in loop 4, are also depicted. The backbone root mean square deviation (RMSD) of the structures in these ensembles relative to the average is 0.4 Å for both K1 and K2 (for the most ordered regions of each protein, which include Cys2-Cys31 in K1 and Cys44-Cys71 in K2). (**C,D**) Surface representations of energy-minimized structures of K1 (**C**) and K2 (**D**). Hydrophobic, basic and acidic residues are colored in green, blue and red, respectively.

The following figure supplements are available for figure 1:

*Figure 1 continued on next page*

*Figure 1 continued*

**Figure supplement 1.** NMR spectra of K1, K2 and DkTx in solution.

**Figure supplement 2.** Backbone assignments of DkTx in solution.

**Figure supplement 3.** Summary of $^3J_{HNH\alpha}$ coupling constant and proton chemical shifts of DkTx in solution.

**Figure supplement 4.** NMR and refinement statistics of K1 and K2

indicates that the structures are determined without significant stereochemical violations. A superposition of the 20 energy-minimized structures obtained for each lobe demonstrates that the protein backbone is well defined (*Figure 1A,B*), except in the inter-connecting linker, for which we obtained relatively few distance restraints. Each lobe consists of two anti-parallel beta-strands (F21-Y22/K32-K33 in K1 and L61-D62/Y70-C71 in K2), stabilized by three disulfide bonds.

Comparison of the structures of K1 and K2 reveals that their fold is similar, but not identical; the root mean square difference (RMSD) in their backbone, for the core 28 residues, is 1.45 Å (1.56 Å including all heavy-atoms). Two notable differences between K1 and K2 are the length of loop 3, which is longer in K1 than in K2, and the configuration of a tryptophan side-chain in loop 2 (W11 in K1 and W53 in K2), which is more solvent accessible in K1 (154 ± 16 Å²) than in K2 (132 ± 7 Å²). The latter stems from actual differences in the NOESY spectra for K1 and K2; the number of NOE cross-peaks between the conserved Trp in loop 2 and the conserved Ile and Pro residues in loop 4 are fewer in K1 than in K2. The likely cause of this structural difference is a serine (S10) to glycine (G52) substitution at the first position in loop 2, which seems to allow this loop to bend towards loop 4 in K2 (*Figure 1A,B*).

Surface renderings of K1 and K2 show that both lobes feature a large cluster of exposed hydrophobic residues (green), comprised of W11, M25, F27 and I28 in K1 and W53, L65, A66, F67 and I68 in K2 (*Figure 1C,D*). These extensive and well-conserved hydrophobic surfaces are consistent with our previous finding that detergents are critical for efficient refolding of the toxin in vitro, and with the observation that folded DkTx is more hydrophobic than the linear form (*Bae et al., 2012*). The hydrophobic surfaces in both K1 and K2 are surrounded by basic residues (blue) and acidic residues (red), including D1, E7, K13, K14, H30 and K32 in K1 and E47, E49, K56, E72, K73 and R75 in K2. This pronounced amphipathic character is reminiscent of well-characterized voltage-sensor toxins such as hanatoxin (*Takahashi et al., 2000*), SGTx1 (*Lee et al., 2004*), VSTx1 (*Jung et al., 2005*) and GxTx-1E (*Lee et al., 2010*), which interact with voltage-sensing domains of Kv channels within a membrane environment (*Gupta et al., 2015*; *Lee and MacKinnon, 2004*; *Milescu et al., 2009*; *Milescu et al., 2007*; *Phillips et al., 2005b*).

## Improved atomic models of TRPV1 with and without DkTx bound

To begin to discern the mode in which DkTx recognizes and activates TRPV1, we set out to improve the existing atomic models of the structure of the channel, both in the apo state as well as in complex with DkTx, based on cryo-EM data obtained in a previous study (*Cao et al., 2013*). Notwithstanding the groundbreaking insights provided by these cryo-EM maps, the associated structural models deposited in the Protein Data Bank (PDB) are objectively suboptimal in several respects. Aside from the missing backbone fragments and side-chains in unresolved regions, the published structural models show a significant number of stereochemical violations and steric overlaps (*Tables 1*, *2*). As an example, the MolProbity score, which is a standardized metric of model quality among structures of comparable resolution, ranks PDB entries 3J5Q and 3J5P in the 29th and 36th percentile, respectively (with 100th percentile being the best score). A specific issue pertaining to the published model of the TRPV1/DkTx/RTx complex is that it is based on a map with imposed fourfold symmetry, despite the fact that the two DkTx lobes are not identical and thus the appropriate symmetry for the complex would be twofold. More importantly, in the published structure of the TRPV1/DkTx/RTx complex, the toxin is modeled as a poly-alanine chain based on the structure of hanatoxin because an atomic-resolution structure of DkTx had not yet been solved. Assignment of

**Table 1.** Evaluation of structural models of apo TRPV1.

**PDB entry 3J5P**

Number of atoms: 18,628

Correlation coefficient with cryo-EM map EMD-5778 (3.4 Å): 0.71

| All-atom contacts | Clashscore, all atoms[a] | 76.31 | | 16th percentile[b] |
|---|---|---|---|---|
| Protein geometry | Poor rotamers | 524 | 26.57% | Ideal: <1% |
| | Ramachandran outliers | 0 | 0.00% | Ideal: <0.05% |
| | Ramachandran favored | 2204 | 94.35% | Ideal: >98% |
| | MolProbity score[c] | 3.82 | | 36th percentile[b] |
| | $C_\beta$ deviations >0.25 Å | 0 | 0.00% | Ideal: 0 |
| | Bad backbone bonds | 166/19032 | 0.87% | Ideal: 0% |
| | Bad backbone angles | 72/25760 | 0.28% | Ideal: < 0.1% |

**Partial ROSETTA model[d]**

Number of atoms: 18,628

Correlation coefficient with cryo-EM map EMD-5778 (3.4 Å): 0.71

| All-atom contacts | Clashscore, all atoms[a] | 3.45 | | 97th percentile[b] |
|---|---|---|---|---|
| Protein geometry | Poor rotamers | 4 | 0.20% | Ideal: <1% |
| | Ramachandran outliers | 12 | 0.51% | Ideal: <0.05% |
| | Ramachandran favored | 2200 | 94.18% | Ideal: >98% |
| | MolProbity score[c] | 1.53 | | 94th percentile[b] |
| | $C_\beta$ deviations >0.25 Å | 0 | 0.00% | Ideal: 0 |
| | Bad backbone bonds | 44/19024 | 0.23% | Ideal: 0% |
| | Bad backbone angles | 56/25744 | 0.22% | Ideal: < 0.1% |

**Complete ROSETTA model**

Number of atoms: 19,632

Correlation coefficient with cryo-EM map EMD-5778 (3.4 Å): 0.71

| All-atom contacts | Clashscore, all atoms[a] | 3.77 | | 96th percentile[b] |
|---|---|---|---|---|
| Protein geometry | Poor rotamers | 8 | 0.37% | Ideal: <1% |
| | Ramachandran outliers | 32 | 1.33% | Ideal: <0.05% |
| | Ramachandran favored | 2236 | 92.70% | Ideal: >98% |
| | MolProbity score[c] | 1.62 | | 92nd percentile[b] |
| | $C_\beta$ deviations >0.25 Å | 0 | 0.00% | Ideal: 0 |
| | Bad backbone bonds | 52/20068 | 0.26% | Ideal: 0% |
| | Bad backbone angles | 148/27144 | 0.55% | Ideal: < 0.1% |

[a]Clashscore is the number of serious steric overlaps (>0.4 Å) per 1,000 atoms.

[b]100th percentile is the best score among structures of comparable resolution; 0th percentile is the worst score. For clashscore, the comparative set of structures was selected in 2004, for MolProbity score in 2006.

[c]The MolProbity score combines the clashscore, rotamer and Ramachandran evaluations into a single score, normalized to be in the same scale as X-ray resolution.

[d]The partial ROSETTA model is identical to the complete ROSETTA model, except that it includes only the set of atoms resolved in PDB entry 3J5P.

the K1 and K2 knots to the EM map was therefore not possible, and so the nature of their interaction with the channel or the membrane has remained unclear.

To tackle these methodological issues, we used the molecular-modeling software suite ROSETTA (*Leaver-Fay et al., 2011*) (see Methods). This approach enabled us to develop atomic models of apo TRPV1 and the TRPV1-DkTx complex that are considerably improved; the updated MolProbity scores rank these complete models in the 92nd and 80th percentile, respectively, while maintaining

**Table 2.** Evaluation of structural models of the TRPV1-DkTx complex.

**PDB entry 3J5Q**

Number of atoms: 18,244

Correlation coefficient with cryo-EM map EMD-5776 (fourfold symmetric, 3.8 Å): 0.78

| All-atom contacts | Clashscore, all atoms[a] | 101.85 | | 10th percentile[b] |
|---|---|---|---|---|
| Protein geometry | Poor rotamers | 457 | 27.33% | Ideal: <1% |
| | Ramachandran outliers | 20 | 0.82% | Ideal: <0.05% |
| | Ramachandran favored | 2264 | 92.33% | Ideal: >98% |
| | MolProbity score[c] | 4.04 | | 29th percentile[b] |
| | C$_\beta$ deviations >0.25 Å | 0 | 0.00% | Ideal: 0 |
| | Bad backbone bonds | 106/18000 | 0.59% | Ideal: 0% |
| | Bad backbone angles | 50/24444 | 0.20% | Ideal: < 0.1% |

**Partial ROSETTA model[d]**

Number of atoms: 18,244

Correlation coefficient with cryo-EM map relion_ct16_halfSUM_mr (twofold symmetric, 4.3 Å): 0.77

| All-atom contacts | Clashscore, all atoms[a] | 4.06 | | 96th percentile[b] |
|---|---|---|---|---|
| Protein geometry | Poor rotamers | 6 | 0.35% | Ideal: <1% |
| | Ramachandran outliers | 36 | 1.48% | Ideal: <0.05% |
| | Ramachandran favored | 2234 | 91.56% | Ideal: >98% |
| | MolProbity score[c] | 1.69 | | 89th percentile[b] |
| | C$_\beta$ deviations >0.25 Å | 4 | 0.17% | Ideal: 0 |
| | Bad backbone bonds | 43/18610 | 0.23% | Ideal: 0% |
| | Bad backbone angles | 100/25258 | 0.40% | Ideal: < 0.1% |

**Complete ROSETTA model**

Number of atoms: 20,814

Correlation coefficient with cryo-EM map relion_ct16_halfSUM_mr (twofold symmetric, 4.3 Å): 0.76

| All-atom contacts | Clashscore, all atoms[a] | 6.39 | | 89th percentile[b] |
|---|---|---|---|---|
| Protein geometry | Poor rotamers | 12 | 0.53% | Ideal: <1% |
| | Ramachandran outliers | 72 | 2.81% | Ideal: <0.05% |
| | Ramachandran favored | 2284 | 89.29% | Ideal: >98% |
| | MolProbity score[c] | 1.92 | | 80th percentile[b] |
| | C$_\beta$ deviations >0.25 Å | 10 | 0.41% | Ideal: 0 |
| | Bad backbone bonds | 150/21288 | 0.70% | Ideal: 0% |
| | Bad backbone angles | 261/28792 | 0.91% | Ideal: < 0.1% |

[a]Clashscore is the number of serious steric overlaps (>0.4 Å) per 1,000 atoms.

[b]100th percentile is the best score among structures of comparable resolution; 0th percentile is the worst score. For clashscore, the comparative set of structures was selected in 2004, for MolProbity score in 2006.

[c]The MolProbity score combines the clashscore, rotamer and Ramachandran evaluations into a single score, normalized to be in the same scale as X-ray resolution.

[d]The partial ROSETTA model is identical to the complete ROSETTA model, except that it includes only the set of atoms resolved in PDB entry 3J5Q.

the original quality of the fit to the experimental cryo-EM maps (**Tables 1**, **2**). To construct the model of the TRPV1-DkTx complex, we used an unpublished cryo-EM map with imposed twofold symmetry, kindly provided by Yifan Cheng and colleagues. Although the resolution of this map is somewhat inferior to that of the fourfold symmetrized map utilized originally (4.3 Å compared to 3.8 Å), densities for each of the four DxTk knots (two per toxin) are clearly discernable. We initially docked the NMR structures of K1 and K2 into these densities using Xplor-NIH, in either a clockwise (CW) or

counter-clockwise (CCW) configuration (viewed from the extracellular side), optimizing the fit of each lobe to the EM map while also applying NOE distance restraints within each knot (based on the 2D $^1$H-$^1$H NMR experiments for K1 and K2 in solution). These two partial models were then input into ROSETTA, which was then used to generate a randomized ensemble of 100 models of the complete TRPV1-DkTx complex for each K1-K2 configuration. Analysis of these two ensembles shows that the majority of the 100 CCW models fit to the experimental cryo-EM map significantly better, and also have a better ROSETTA score, than the majority of the 100 CW models (*Figure 2E*). This result is consistent with the observation that the number of NOE-restraint violations in the CW initial seed model generated with Xplor-NIH was twice as large as for the CCW model, for a comparable fit quality (*Figure 2—figure supplement 1*). We can therefore conclude unambiguously that DkTx binds to the outer pore of TRPV1 in a CCW configuration (*Figure 2B,D,F*; *Video 1*).

Further improvements of the ROSETTA models for apo and DkTx-bound TRPV1 focused on the configuration of the side-chains poorly resolved in the EM map (approximately 80% of the side-chain atoms). An ensemble of 12,000 models, weighted by the ROSETTA energy function, were generated in each case, and the most representative among these were identified through a clustering analysis. The selected models for apo and DkTx-bound TRPV1 further improve the ROSETTA score without compromising the quality of the fit to the experimental cryo-EM maps (*Figure 2—figure supplement 2*). Relative to the structures deposited in the PDB, these optimized models also showed a generalized improvement in ProQM per-residue scores (*Ray et al., 2010*) (*Figure 2—figure supplement 3*), in addition to the improved MolProbity global scores mentioned above (*Tables 1*, *2*). Both models are publicly available upon request to the authors.

## Interactions of DkTx with TRPV1 and the surrounding membrane

The model of the TRPV1-DkTx complex reveals that K1 and K2 engage the outer pore of TRPV1 using loop 2 and loop 4 (*Figure 3A,B*). These two loops straddle the interface between TRPV1 subunits, with loop 2 capping the S6 helix from one subunit, while loop 4 localizes near the N-terminus of the re-entrant pore helix from the adjacent subunit. Scanning mutagenesis previously identified four residues within the outer pore domain of TRPV1, namely I599A at the top of S5, F649A in the pore loop, and both A657P and F659A at the top of S6, where mutations disrupt activation of the channel by DkTx (*Bohlen et al., 2010*); two additional mutants in this region (V595A in S5, and T650A in the pore loop) also display diminished activation by the toxin, whereas Y631A, near the N-terminus of the pore helix, enhances the effect of DkTx. Our model indicates that among these residues, only F649, A657, T650 and Y631 could interact directly with the toxin, suggesting that the other mutations influence toxin activation of the channel through an indirect mechanism (see below). Interestingly, the model also shows that both K1 and K2 drape over the top of S6 at the presumed interface with the surrounding membrane, and position several conserved hydrophobic residues (W11, F27 and I28 in K1, and W53, F67 and I68 in K2) where they would interact directly with lipids in the bilayer.

Previous studies have shown that both K1 and K2 can activate TRPV1, but that responses to K1 are much weaker than K2 (*Bae et al., 2012*; *Bohlen et al., 2010*). In the case of the K2 knot, the concentration-dependence for activation of TRPV1 saturates in the low micromolar range, and the extent of maximal activation is comparable to saturating concentrations of DkTx. In contrast, saturation of the concentration-response relation cannot be observed for K1, and at the highest concentrations that can be tested, K1 produces between 6- and 12-fold weaker activation compared to a saturating concentration of K2 (see *Figure 3F*). Because we cannot achieve higher concentrations of K1 in aqueous solution due to limited solubility, we investigated whether this knot has lower affinity or efficacy compared to K2 by taking advantage of the fact that bivalency increases the local concentration of each lobe, and therefore will produce higher occupancy of the channel compared to the separate lobes. For these experiments, we produced bivalent versions of K1 (K1K1) and K2 (K2K2), and tested whether bivalency altered the large difference in the ability of the two lobes to activate TRPV1. K1K1 should remain a weak activator of TRPV1 if K1 has lower efficacy than K2, however, bivalency should diminish the differences between K1 and K2 if K1 has lower affinity. Indeed, we found that 1 μM K1K1 produced comparable activation to that produced by the same concentration of DkTx or K2K2 (*Figure 3C,D*), suggesting that the efficacy of the two knots is comparable, but that the affinity of K1 is much lower than that of K2. Although both DkTx and K2K2 exhibited slow

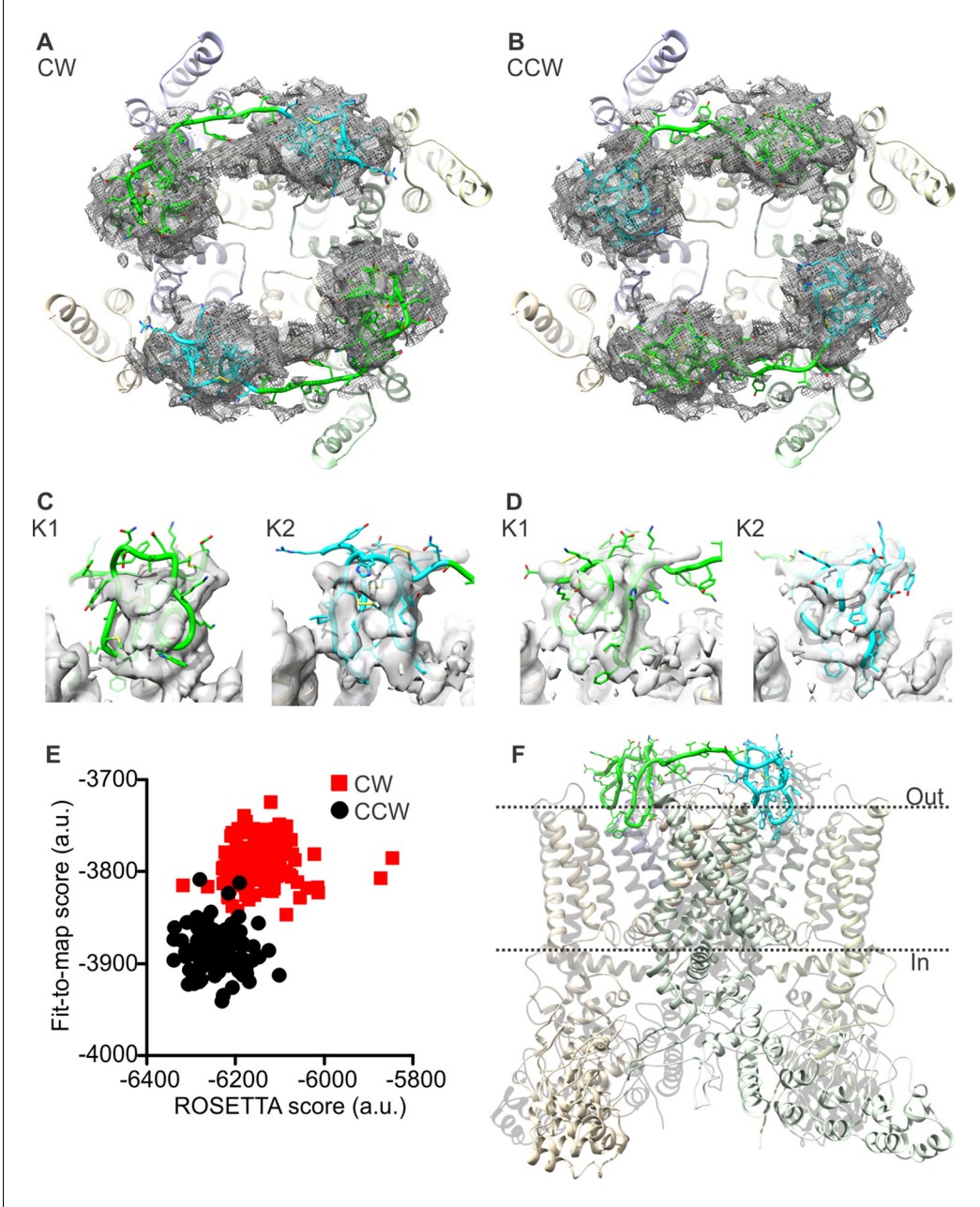

**Figure 2.** Docking of DkTx into the electron density map of DkTx/RTx-bound TRPV1. (**A,B**) Top-ranking ROSETTA models of the structure of TRPV1 with DkTx bound in a clockwise (CW, **A**) or counter-clockwise (CCW, **B**) configuration. The structures are overlaid on the experimental twofold symmetric cryo-EM map (unpublished data kindly provided by Yifan Cheng and colleagues), and densities close to the toxin molecules are shown at a contour level of 3σ. K1 and K2 are shown in green and cyan, respectively. (**C,D**) View of K1 and K2 from the membrane plane, either in the CW (**C**) or CCW (**D**) configuration. The cryo-EM map is shown as a gray surface. (**E**) Quantitative evaluation of the CW and CCW configurations of DkTx bound to TRPV1 using ROSETTA. The ROSETTA fit-to-density score and the conventional ROSETTA score (excluding the fit-to-density contribution) are reported for 100 models generated for either the CW or CCW configuration. (**F**) View of the top-ranking ROSETTA model of the TRPV1-DkTx complex, with the K1 and K2 in a CCW configuration.

The following figure supplements are available for figure 2:

**Figure supplement 1.** Evaluation of the CW and CCW orientations for docking of K1 and K2 to TRPV1 using Xplor-NIH.

*Figure 2 continued on next page*

*Figure 2 continued*

**Figure supplement 2.** Refinement of cryo-EM structures of apo and DkTx-bound TRPV1 using ROSETTA.
**Figure supplement 3.** Improvements in the newly refined structural models of apo and DkTx-bound TRPV1.

---

dissociation, we observed considerably more rapid dissociation for K1K1 (*Figure 3C*), confirming that K1 has lower binding affinity than K2 even when tested in its bivalent form.

The observation that K1 has lower affinity than K2 provides a means to validate our structural model of the TRPV1-DkTx complex. Specifically, chimeras in which we replace either loop 2 and loop 4 in K1 by that in K2 ought to lead to a gain of function for activating the channel, resulting from an enhanced binding affinity; in contrast, substitutions in regions not involved in the channel-toxin interface should not alter the degree of channel activation. We generated such chimeras between K1 without the linker (K1-ΔL) and K2 (*Figure 3E*), and tested their ability to activate full-length TRPV1 channels compared to a saturating concentration of capsaicin (*Figure 3F*). The weak activation observed with K1-ΔL (*Figure 3F*; green) was not altered in constructs in which either Asn1 in the N-terminus or loop 3 of K2 were transferred into K1-ΔL [i.e. K1(D1N) or K1(K2L3)] (*Figure 3F*). In contrast, chimeras in which loop 2, loop 4 or the C-terminal part of K2 were transferred into K1-ΔL [i.e. K1(K2L2), K1(K2L4) or K1(K2CT)] exhibited considerably more robust activation compared to K1-ΔL (*Figure 3F*). The negative results observed with the N-terminal mutant and the loop 3 transfer, together with the gain of function observed with loop 2 and loop 4 transfer, support the notion that these two loops engage directly with the outer pore of TRPV1, as observed in our model (*Figure 3G*). The model also provides a possible explanation for the gain of function observed when the C-terminus of K2 is transferred to K1. The location of the C-termini of K1 and K2 indicates that they are likely to interact with the pore turret (residues 604–626) of TRPV1 (*Figure 3G*), a region absent in our model as it was deleted from the TRPV1 construct used for cryo-EM (*Liao et al., 2013*), but preserved in the full-length construct used for our functional studies.

To investigate the interaction between the toxin and channel in more detail, we carried out an all-atom MD simulation of the complex embedded in a phospholipid bilayer (*Figure 4A*). To enhance the exploration of diverse interaction patterns in a limited simulation time (~500 ns), we coupled the $\chi_1$ and $\chi_2$ torsion angles of all interfacial side-chains in the toxin and channel to a fictitious high-temperature bath, using an extended-Lagrangian approach (*Iannuzzi et al., 2003*) (see Methods). To preclude the dissociation of the complex under this bias, the cryo-EM envelop was used as a three-dimensional restraint. A contact map between residues in DkTx and the outer pore of TRPV1 was then generated from the last 200 ns of simulation, so as to identify the most pronounced interactions (*Figure 4D*). Two regions of the extracellular surface of the channel stand out as forming the most persistent side-chain contacts with either of the two lobes of DkTx, namely the N-terminus of the pore-helix, primarily via Y631, and a stretch of the pore-loop and N-terminus of S6, including N652, D654, F655, K656, A657 and V658; additional interactions are mediated by K535 and E536, in S4 (*Figure 4D*; *Figure 4—figure supplement 1*; *Video 2*). The contacts with the pore and S6 helices are particularly worth noting because the A657P mutation effectively abolishes channel activation by DkTx, while Y631A enhances it (*Bohlen et al., 2010*). Many of the contacts on the toxin are with residues that are equivalent in

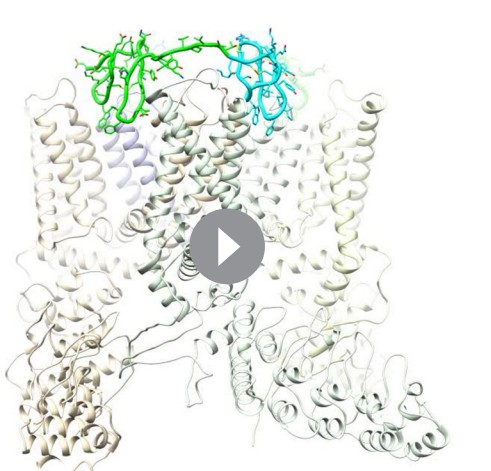

**Video 1.** The top-ranking ROSETTA model of the TRPV1-DkTx complex, with DkTx in a CCW configuration. The K1 knot and linker is colored green and the K2 knot is colored blue.

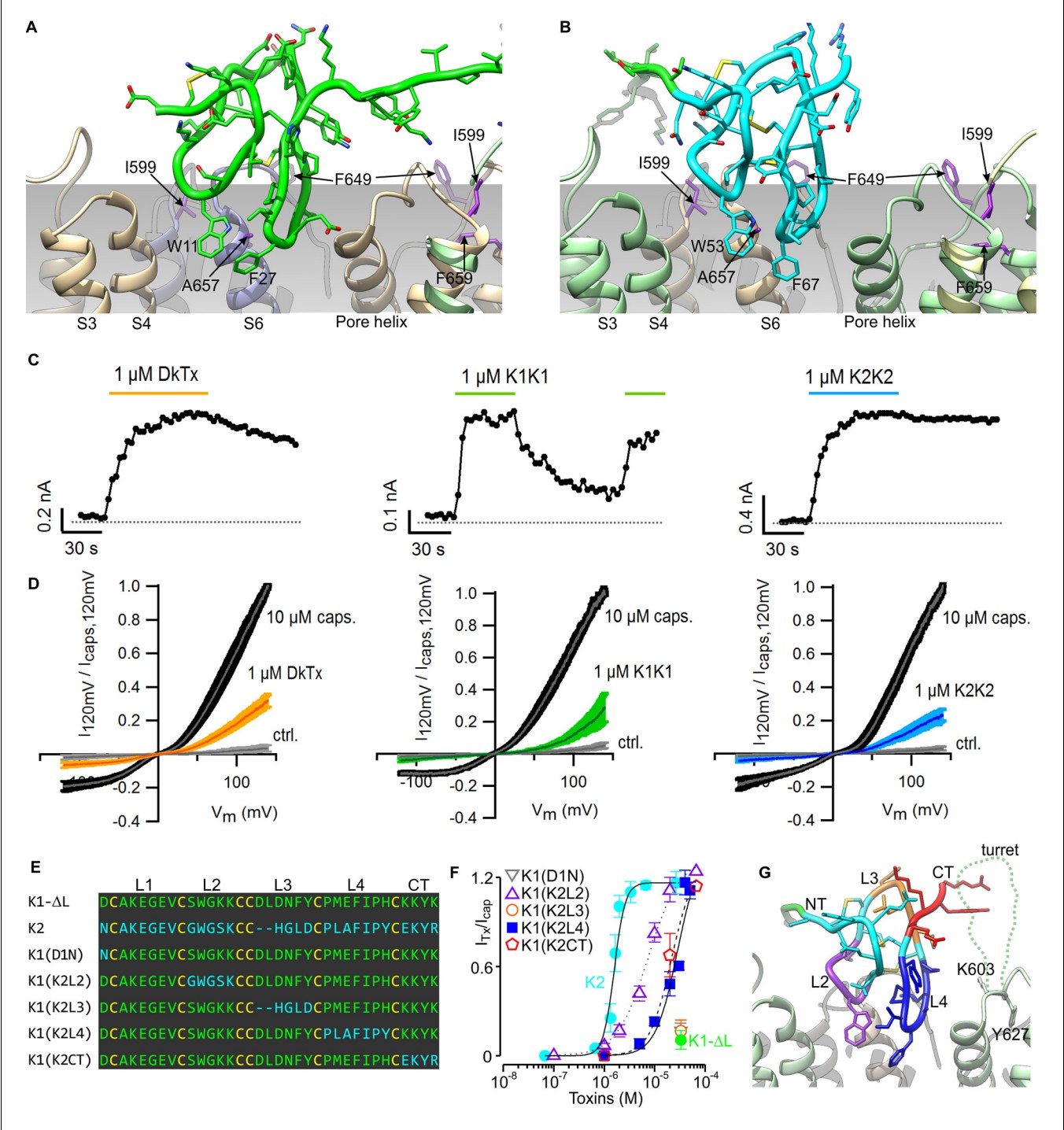

**Figure 3.** Validation of the proposed structural model of the TRPV1-DkTx complex. Close-up of K1 (**A**) and K2 (**B**) in the top-ranking ROSETTA model of DkTx bound to TRPV1. Side chains of TRPV1 residues where mutation disrupts DkTx-induced channel activation (I599, F649, A657 and F659) (*Bohlen et al., 2010*) are highlighted in purple. (**C**) Representative time courses of TRPV1 activation at +120 mV by either DkTx (left panel), K1K1 (middle panel) or K2K2 (right panel) measured from a train of voltage-ramps in the whole-cell configuration with the TRPV1 expressed in HEK cells. In the case of K1K1, toxin was reapplied to verify that the observed decrease in current upon removal of the toxin was due to toxin dissociation from the channel rather than desensitization. Voltage ramps were from −120 to +140 mV over 1 s and were applied every 3 s from a holding potential of −90 mV. The colored horizontal lines denote the application of toxins. The dotted gray line denotes the zero-current level. Fractional dissociation 1 min after toxin removal was quantified across cells as the current amplitude 1 min after toxin removal divided by the steady-state current amplitude in the presence of the toxin. Fractional dissociation values (mean ± sem) were 0.8 ± 0.05 (n = 8) for DkTx, 0.3 ± 0.04 (n = 5) for K1K1 and 0.96 ± 0.03 for K2K2

Figure 3 continued

(n = 6). (D) Mean normalized I-V relations obtained from voltage ramps at steady-state for control solution (gray), toxins (yellow, green or blue) and saturating capsaicin (10 µM; black) in the whole-cell configuration using transiently transfected HEK293 cells. The thin colored curves represent the mean and the thicker envelope the standard error (n = 5–6 for each panel). (E) Primary sequence of K1 without linker (K1-ΔL), K2 and gain-of-function chimeras between K1 and K2. Residues of K1-ΔL and K2 are shown in green and cyan, respectively, and cysteines are shown in yellow. Loops between cysteine residues are labeled as L1, L2, L3 or L4. CT denotes C-terminal region. (F) Concentration-dependence for activation of TRPV1 by K1, K2 and chimeras in two-electrode voltage clamp recordings of oocytes expressing TRPV1. Activation of the full-length TRPV1 channel by the toxin at different concentrations was measured in oocytes at a holding voltage of −60 mV. Toxin-induced currents ($I_{Tx}$) were normalized against the current activated by a saturating concentration of capsaicin ($I_{cap}$ at 10 µM), in the same cell. Note that the data for K1(D1N) and K1(K2L3) are obscured by the data for K1-Δ L. Error bars represent SEM (n = 3). (G) Close-up of the structure of K2 bound to TRPV1 with regions of the toxin colored with the same scheme as in (F). Side chains of K603 and Y627 indicate the region where the pore turret (residues 604–626) would likely reside if not deleted in the construct used for cryo-EM and in our model.

K1 and K2, e.g. W11 and W53, F27 and F67, K14 and K56 and G12 and G54, respectively, and a computational alanine-scanning (Ala-scan) analysis of the toxin-channel interface, based on the configurations explored during the MD simulation (see Methods), indicate that these are all influential (*Figure 4—figure supplement 2*). However, there are also differences between K1 and K2, which might underlie their different affinity for TRPV1. The most interesting unique interactions for K2 involve S55 (K13 in K1), L65 (M25 in K1) and R75 (K35 in K1). The persistence of these interactions is worth noting because these residues are located in either loop 2 (S55), loop 4 (L65) or the C-terminus (R75), and transfer of these segments from K2 into K1 yields a pronounced increase in the binding affinity of isolated K1 (*Figure 3F*). These residues are also predicted to have a significant stabilizing effect by the computational Ala-scan (*Figure 4—figure supplement 2*).

Interestingly, the simulation of the TRPV1-DkTx complex also reveals that interactions between toxin and the membrane occur concurrently with those with the channel (*Figure 4B,C,E*). Several hydrophobic residues in loop 2 and loop 4 of both K1 and K2 stand out as forming long-lived interactions with lipids, including W11, F27 and I28 in K1, and W53, F67 and I68 in K2. Both knots also showed some unique interactions with lipid head-groups, for example Y70 (in K2) and H30 (in K1) (*Figure 4B,C,E*). In summary, therefore, the simulation indicates that K1 and K2 interact similarly with the outer pore of TRPV1 and the surrounding membrane, but also reveals a number of protein-protein and protein-lipid interactions that might explain the higher affinity of K2 for the TRPV1 channel.

## Membrane partitioning of DkTx

Motivated by the amphipathic character evident in the structure of DkTx, together with the placement of the toxin at the protein-lipid interface when bound to TRPV1, we investigated the interaction of DkTx and each of its two lobes with lipid membranes. For these experiments, we took advantage of the presence of a single conserved and solvent accessible Trp residue in loop 2 of both K1 and K2 (*Figure 1C,D*; W11 and W53), and used Trp fluorescence to monitor the partitioning of the toxin from the aqueous solution into the membrane environment (*Gupta et al., 2015*; *Ladokhin et al., 2000*; *Milescu et al., 2007*). When aqueous solutions of DkTx were excited at a wavelength of 280 nm, emission spectra with maxima at 353 nm were obtained (*Figure 5A*, left), as expected for Trp residues residing in such an environment. In contrast, upon addition of large unilamellar vesicles (LUVs), the fluorescence emission spectra for DkTx shifted to shorter wavelengths (i.e. a blue shift), and the fluorescence intensity increased, suggesting that the toxin partitions into membranes where the environment of the Trp side-chains is more hydrophobic and their dynamics are more constrained (*Ladokhin et al., 2000*). To quantify the extent of partitioning as a function of lipid concentration, we measured the relative fluorescence intensity in a blue-shifted region of the spectra (e.g. 320 nm) as a function of lipid concentration, and fit a partition function to the data to obtain a mole-fraction partition coefficient ($K_x$) of $(2.3 \pm 0.7) \times 10^6$, indicating an energetically favorable interaction of DkTx with lipid membranes (*Figure 5A*, right). This strong interaction of DkTx with membranes would facilitate binding of the toxin to the channel by increasing the local concentration of the toxin near the channel and by a reduction in the dimensionality of diffusion within the membrane (*Axelrod and Wang, 1994*).

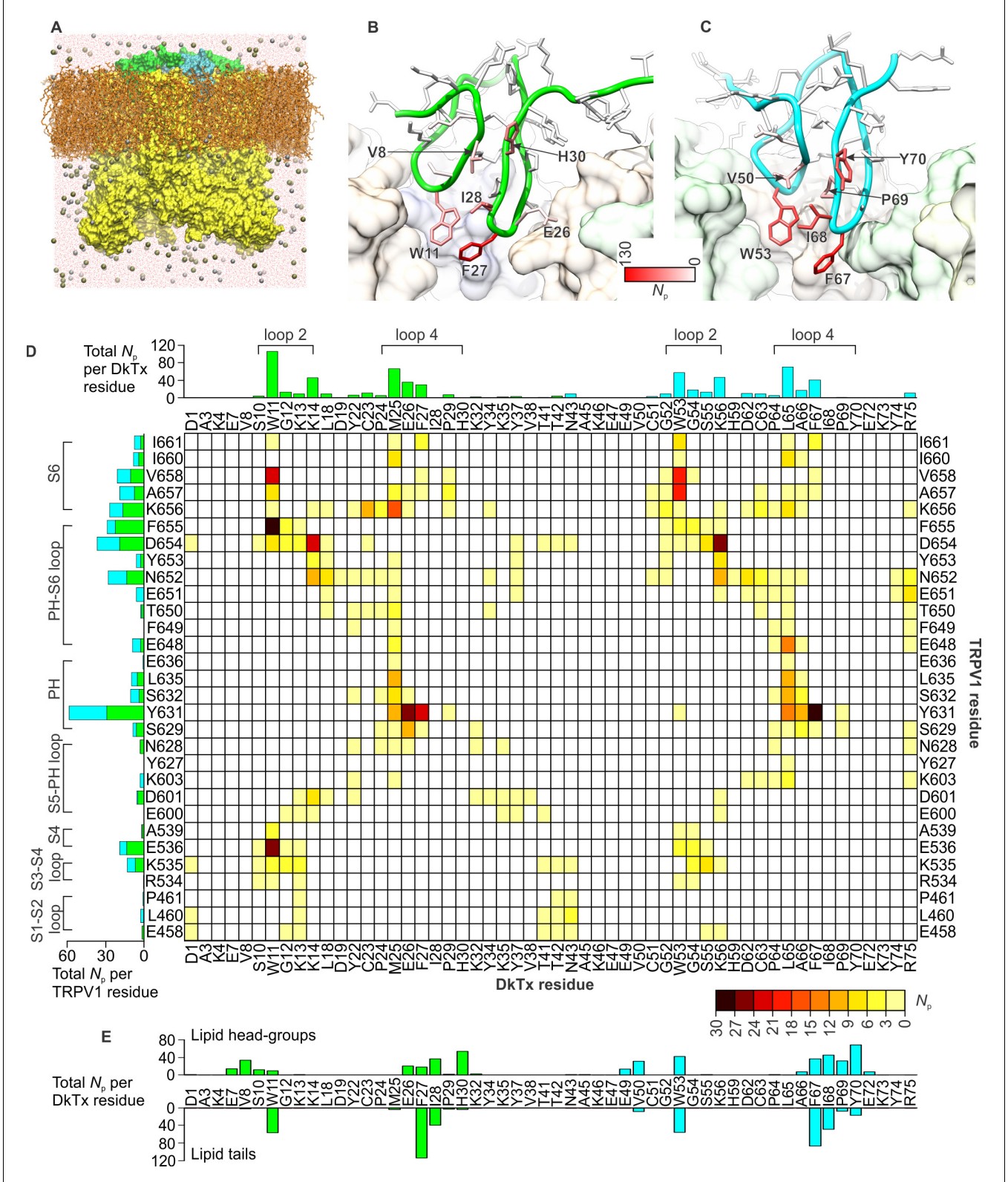

**Figure 4.** DkTx interactions with TRPV1 and the surrounding lipid bilayer from MD simulations. (**A**) Molecular system employed in the simulation of the TRPV1-DkTx complex. The molecular surface of the channel is shown in yellow, while the two bound DkTx molecules are shown in green (K1) and cyan (K2). Electrolyte ions are shown as silver (Na⁺) and tan (Cl⁻) spheres, while water molecules are indicated by red dots, and the POPC bilayer in orange. A comparable system was used to simulate the apo channel. (**B, C**) Residues of K1 (**B**) and K2 (**C**) that continue to interact with the membrane after DkTx

*Figure 4 continued on next page*

*Figure 4 continued*

recognizes TRPV1. The side-chains in the toxin are color-coded according to the number of contacting atom pairs, $N_p$, that were observed for each of these side-chains and any lipid molecule in the membrane (see Methods). The values plotted are averages over the 200 ns accelerated MD simulation of the TRPV1-DkTx complex. (D) Analysis of the side-chain contacts between TRPV1 and DkTx, as observed in the accelerated MD simulation of the complex. The color-coded matrix reports on the contacts between specific side-chain pairs in the toxin and channel. The color-key reflects the number of contacting atom pairs in each case, $N_p$, averaged over the simulated time (see Methods). The bar charts report on the total value of $N_p$ for each residue in either the toxin (top) or the channel (left), when any residue in the other protein is considered. The data for K1 and K2 are colored differently (green and cyan, respectively) in the bar graphs. (E) Contacts between DkTx residues and lipid molecules in the surrounding membrane. The bar plot reflects the number of contacting atom pairs between any of the lipid head-groups or tails, and each toxin residue.

The following figure supplements are available for figure 4:

**Figure supplement 1.** Interactions between DkTx and TRPV1 from MD simulations.

**Figure supplement 2.** Computational alanine-scan of the DkTx-TRPV1 interface.

We conducted similar experiments for K1 and K2 separately, and observed that both K1 and K2 exhibit blue shifts upon addition of LUVs, although interestingly the blue shift of K2 was considerably smaller compared to K1 (*Figure 5B,C*). Fitting of a partition function to the data yielded $K_x$ values of $(3.9 \pm 1.1) \times 10^5$ for K1 and $(2.0 \pm 0.2) \times 10^4$ for K2, suggesting that the interaction of K1 with lipid membranes is stronger compared to K2. Because the fluorescence changes for K2 were small and did not saturate even at higher lipid concentrations (*Figure 5C*), we conducted a similar experiment using the bivalent constructs K1K1 and K2K2, which we expected would increase the $K_x$ values due to avidity (*Figure 5D,E*). Indeed, both K1K1 ([$2.9 \pm 0.3] \times 10^6$) and K2K2 ([$8.1 \pm 2] \times 10^4$) showed increased $K_x$ values compared to K1 and K2, respectively, and K1K1 exhibited a higher $K_x$ value when compared to K2K2, confirming that partitioning of K1 into membranes is more favorable than K2. These results are remarkable when considering that K2 has a higher affinity for TRPV1 compared to K1, leading us to propose a model whereby partitioning of DkTx into membranes would be disproportionately mediated by K1, leaving K2 available to engage TRPV1 and initiate formation of the toxin-channel complex (see Discussion).

We also examined chimeras of K1 and K2 to identify which regions determine their distinct membrane partitioning energies. Interestingly, transfer of the N-terminus and loops 2 and 4 from K2 into K1 produced either no change, or a moderate increase in $K_x$, even though the region transferred was from the weaker partitioning K2 knot (*Figure 5—figure supplement 1*). It is possible that the C-terminus of K2 is responsible for the weaker partitioning of that knot; however, transfer of that region into K1 greatly diminished the blue-shifts observed on addition of membrane vesicles (*Figure 5—figure supplement 1E*), precluding an accurate determination of $K_x$.

Although bivalency in DkTx clearly plays an important role in channel activation, it also increases the local concentration of the two lobes relative to each other and may promote interactions between the two. Indeed, the measured free energies of membrane partitioning ($\Delta G° = -RT \ln K_x$) for DkTx ($\Delta G° = -8.5$ kcal mol$^{-1}$), K1K1 ($\Delta G° = -8.7$ kcal mol$^{-1}$) and K2K2 ($\Delta G° = -6.6$ kcal mol$^{-1}$) reveal a systematic energetic penalty for bivalency between 4.8 to 6.3 kcal mol$^{-1}$ if we calculate the theoretical free energies for bivalent toxins (DkTx $\Delta G° = -13.3$ kcal mol$^{-1}$; K1-K1 $\Delta G° = -15$ kcal mol$^{-1}$; K2-K2 $\Delta G° = -11.6$ kcal mol$^{-1}$) from that measured for the monovalent toxins (K1 $\Delta G° = -7.5$ kcal mol$^{-1}$; K2 $\Delta G° = -5.8$ kcal mol$^{-1}$) assuming

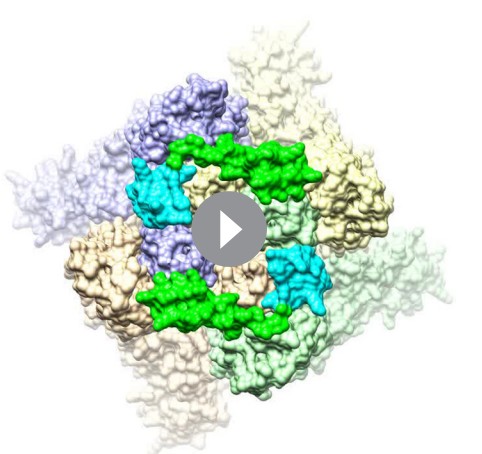

**Video 2.** Close up of the interface between TRPV1 and DkTx, color-coded according to the persistence of protein-protein contacts from MD simulations. See also *Figure 4—Supplement 1*.

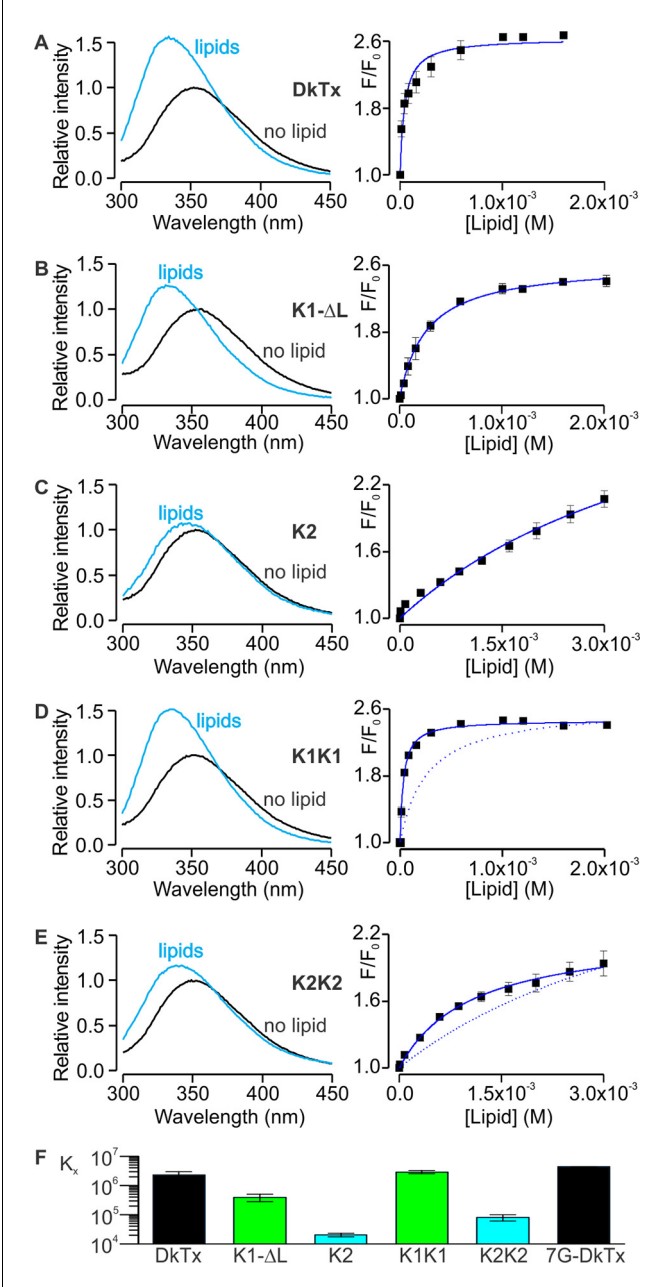

**Figure 5.** Interaction of DkTx and toxin constructs with lipid membranes measured using Trp fluorescence. (A–E, left panel) Trp emission spectra of DkTx (**A**), K1 without linker (**B**), K2 (**C**), bivalent K1 (K1K1) (**D**) and bivalent K2 (K2K2) (**E**) in the absence (black) and presence (blue, 1.6 mM for DkTx, 2 mM for K1-ΔL and K1K1, and 3 mM for K2 and K2K2) of lipid vesicles (1:1 mix of POPC:POPG). (A–E, right panel) Relative fluorescence intensity at 320 nm ($F/F_0$) as a function of available lipid concentrations (60% of total lipid concentration). Smooth curves correspond to partition functions with $K_x = (2.3 \pm 0.7) \times 10^6$ and $F/F_0^{max} = 2.65 \pm 0.08$ for DkTx, $K_x = (3.9 \pm 1.1) \times 10^5$ and $F/F_0^{max} = 2.63 \pm 0.07$ for K1-ΔL, $K_x = (2.0 \pm 0.3) \times 10^4$ and $F/F_0^{max} = 3.71 \pm 0.40$ for K2, $K_x = (2.9 \pm 0.3) \times 10^6$ and $F/F_0^{max} = 2.47 \pm 0.03$ for K1K1 and $K_x = (8.1 \pm 2) \times 10^5$ and $F/F_0^{max} = 2.77 \pm 0.55$ for K2K2. Error bars correspond to SEM (n = 3 or 4). Partition functions for K1-ΔL and K2 are shown as dotted lines for comparison in **D** and **E**, respectively. (**F**) Comparisons of mole-fraction partition coefficient ($K_x$) of toxins. 7G-DkTx denotes a construct of DkTx whose linker (PYVPVTT) is replaced with 7 Gly residues. Error bars correspond to SEM (n = 3 or 4).
The following figure supplement is available for figure 5:

**Figure supplement 1.** Interaction of K1-K2 chimeras with lipid membranes measured using Trp fluorescence.

complete additivity of the free energies for partitioning of each lobe. One possible explanation for this energetic penalty is that the linker between the two knots constrains their orientations and prevents optimal membrane interactions by both lobes concurrently. To test this possibility we constructed a version of DkTx wherein the linker was replaced with a highly flexible linker comprised of 7 Gly residues (7G-DkTx), and observed a $K_x$ value that is indistinguishable from wild-type DkTx ([4.4 ± 0.03] × 10$^6$, *Figure 5F*). This result seems to rule out the possibility that the linker restricts the dynamics of the two lobes significantly. Therefore, we deduce that the suboptimal membrane partitioning of DkTx probably owes to direct interactions between the two lobes, possibly mediated through the same amphipathic surfaces that interact with the membrane. Although we could not observe NOEs between K1 and K2 in the NMR spectra of DkTx in solution, it is conceivable that transient and non-specific hydrophobic interactions could go undetected, or that they are prevalent only when the toxin dynamics becomes restricted by the membrane surface.

## Insights into the mechanism of channel activation

As noted previously (*Cao et al., 2013*), the most striking features of the cryo-EM structure of TRPV1 bound to DkTx, relative to the unbound structure, are the dilation of the outer pore within the selectivity filter (SF) region, and the opening of the intracellular gate that is formed at the crossing between the S6 helices. Based on our improved structures of TRPV1 with and without DkTx, we set out to systematically analyze the key differences between these two states to explore the mechanism by which toxin binding promotes channel opening.

This comparative analysis indicates that binding of DkTx promotes changes in the transmembrane architecture of the channel that pertain not only to the internal structure of each of the channel subunits, but also to their relative arrangement (*Figure 6*); as a result, these seemingly cooperative changes open up the constrictions observed in the apo structure, both in the SF region and the intracellular gate. These structural changes appear to be effected through displacements in the pore helix (P) and S6 helices relative to the S1-S4 unit and the transmembrane S5 segment; individually, the S1-S5 and P-S6 units are largely unchanged (RMSD ~ 0.7–0.8 Å) (*Figure 6A–C*). That the P and S6 helices become displaced relative to S1-S5 (*Figure 6D*) is consistent with the fact that their respective N-termini are the primary contacts for the toxin on the channel extracellular surface, as discussed above. The changes in P and S6 within a given subunit correlate very clearly with a pronounced rearrangement of the SF and pore loop in that same subunit (*Figure 6D*), but also with changes in the intracellular side of the channel. The rationale for this remote effect is that the P and S6 helices are the main interfacial elements between adjacent subunits in the transmembrane domain of the channel (*Figure 6E*); therefore, displacements in these elements are propagated to the adjacent subunits, and compounded to the changes in their own internal structure. As a result, the S1-S4 units (which as mentioned remain largely unchanged) become noticeably displaced relative to each other, thus affecting the crossing angle of helices S5 and S6, and hence the degree of opening of the intracellular gate (*Figure 6F*).

Interestingly, the structural changes in the main-chain of the channel that are observed upon DkTx binding correlate with the disruption of a cluster of hydrophobic interactions behind the SF, at the interface between S5, the pore helix and S6, in close proximity to the contact region with the toxin (*Figure 6G,H*; *Video 3*). This network involves, among others, residues I599, F659 and V595, F649, as well as T650. Only the latter two residues are nearby DkTx in the toxin-bound structure; however, mutation of any of these five residues diminishes toxin-induced opening of the channel, particularly in the case of I599A, F649A and F659A (*Bohlen et al., 2010*). To evaluate whether the distinct arrangement of this hydrophobic cluster is a significant feature of the apo and toxin-bound structures, despite their limited resolution, a MD simulation of apo TRPV1 was also carried out and contrasted with that discussed above for the toxin-bound structure (*Figure 4*), using an analogous methodology. When the degree of compactness of this hydrophobic network is quantified in terms of the number and persistence of pairwise side-chain contacts, it is apparent that there is a generalized reduction of these contacts in DkTx-bound TRPV1, relative to apo TRPV1 (*Figure 6I*). In DkTx, two of the residues nearest to this hydrophobic cluster are M25 in K1 and L65 in K2, a relatively conserved position on the two knots that makes frequent contact with channel residues in the MD simulations (*Figure 4D* and *Figure 6G*). Accordingly, both M25 and L65 are predicted to play energetically important roles based on our computational alanine-scan of the toxin-channel interface (*Figure 4—figure supplement 2*). Although we have not yet tested other toxin residues that make

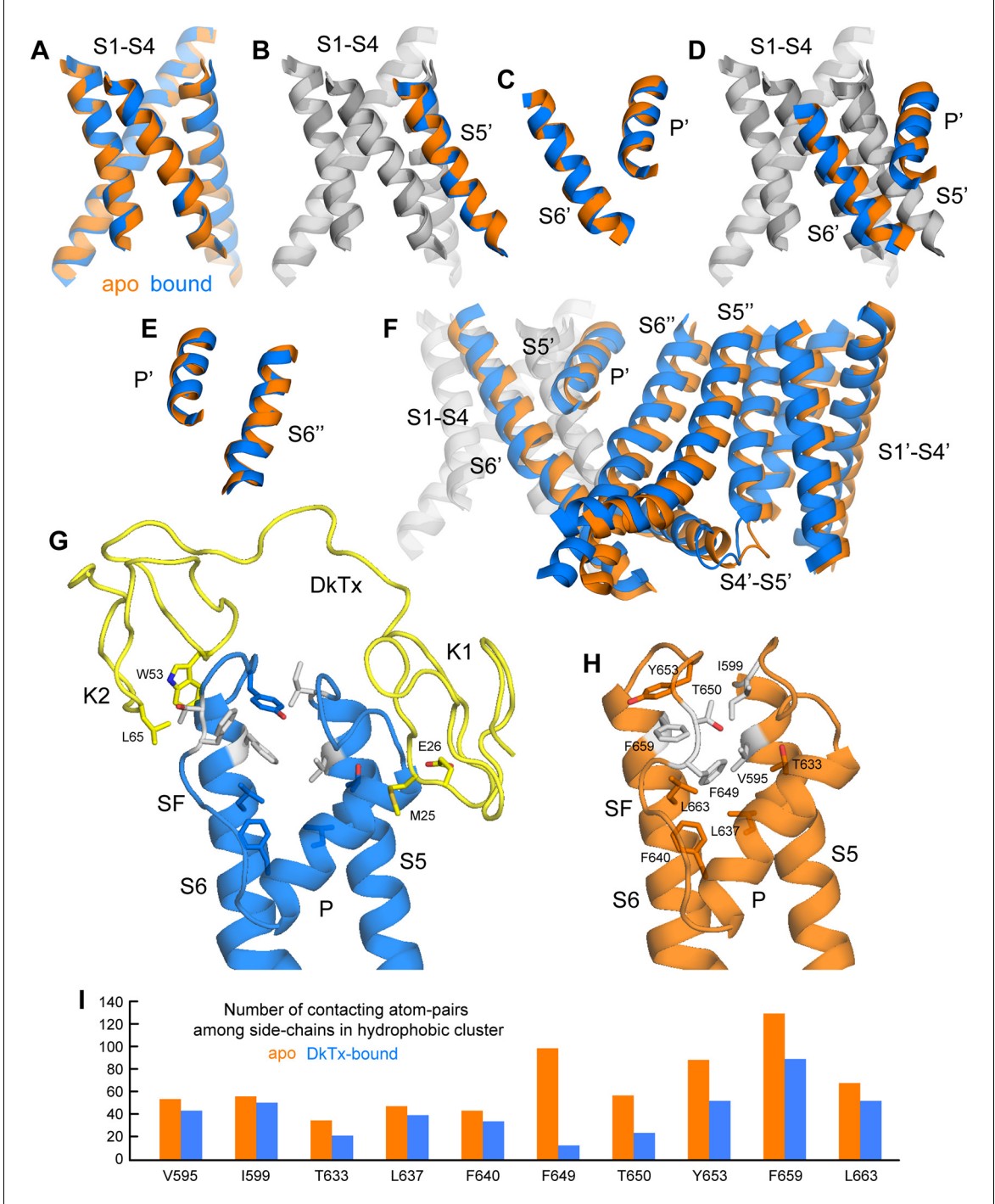

**Figure 6.** Conformational mechanism of activation of TRPV1 inferred from the cryo-EM structures of apo and DkTx-bound channel. (**A**) Overlay of the S1-S4 unit in the apo (orange) and toxin-bound (marine) structure; the root-mean-square difference (RMSD) of the C-α trace is only ~0.7 Å, indicating the internal structure of this unit is not altered upon toxin binding. (**B**) Overlay of the S1-S4 unit, plus the transmembrane portion of the neighboring S5, which belongs to the adjacent protomer (hence referred to as S5'); the RMSD is again ~0.7 Å. (**C**) Overlay of the pore helix (P') and the transmembrane portion of S6'; the RMSD is also ~0.7 Å. (**D**) Overlay of the S1-S4 unit, plus S5', the pore helix P' and S6'. Only the S1-S4 unit and S5' (in gray) are used for fitting, to highlight the relative displacement of S6' and the P' helix, which, as shown in panel (**C**), move as a largely rigid unit. (**E**) Overlay of the pore helix P' and the transmembrane portion of S6 across the protomer interface, i.e. S6"; the RSMD is ~ 0.6 Å. (**F**) Overlay of two adjacent channel subunits, using the same fit as in panel (**D**). Owing to the quasi-rigid structure of the P-S6 interface across subunits (see panel **E**), the displacements induced upon toxin binding on the pore and S6 helices result in a change in the relative orientation of the S1-S4 units, which we propose leads to a change in the crossing-angle of S5 and S6 on the intracellular side, and thus the opening of the lower gate. (**G,H**) Close-up view of the changes induced upon

*Figure 6 continued on next page*

*Figure 6 continued*
toxin (yellow) binding on a hydrophobic cluster formed by side-chains from S5, the pore helix, the selectivity filter (SF) and S6. Mutation of the side-chains colored in gray has been shown to diminish toxin-induced TRPV1 activation (see text). (I) Evaluation of the compactness of the hydrophobic cluster shown in (G,H) during MD simulations of apo and toxin-bound TRPV1, in terms of the number of contacting atom pairs of a given side-chain with all others. Fewer contacts imply a less compact arrangement with greater exposure to the solvent, and potentially, an increased heat-capacity of the channel.

direct contact with TRPV1 (such as W11/W53 and K14/K56), we mutated L65 in K2 to Ala and measured the concentration-dependence for activation of TRPV1 relative to wild-type K2. The L65A mutant dramatically reduced activation of K2 and increased the rate of dissociation (*Figure 7A,B*), confirming this residue mediates important interactions with the outer pore of TRPV1.

## Discussion

The goal of this study was to determine the structure of DkTx and explore its interaction with the TRPV1 channel using the recently reported electron density maps of TRPV1 in complex with the toxin. The solution NMR structure of DkTx indicates that the toxin is composed of two well-ordered ICK lobes, connected by a 7 residue linker (*Figure 1*), as previously surmised (*Cao et al., 2013*). Our structures show that K1 and K2 lobes have similar amphipathic surfaces, which are formed by clusters of solvent-accessible hydrophobic residues (aromatic and aliphatic residues in loop 2 and loop 4) and surrounding basic and acidic residues (*Figure 1*). In an improved model of the toxin docked onto the TRPV1 channel, these amphipathic surfaces on K1 and K2 can be seen to intimately interact with the TRPV1 channel, as well as with lipids in the surrounding membrane (*Figures 3*, *4*). Indeed, conserved aromatic residues in DkTx (W11 and F27 in K1, W53 and F67 in K2) reach into the void formed by S4, S6 and pore-helix of TRPV1, which lipid molecules would fill in the absence of the toxin (*Figures 3*, *4*). Our results demonstrate that DkTx interacts favorably with membranes in the absence of the channel such that the conserved Trp residues reside within the membrane environment and exhibit blue-shifted fluorescent emission spectra (*Figure 5*). Taken together, our results provide direct structural evidence for a model wherein DkTx interacts with TRPV1 within the lipid membrane. Although the bivalency of DkTx clearly helps to prolong the lifetime of the toxin-channel complex (*Bohlen et al., 2010*), each lobe has relatively small protein-protein interfaces with TRPV1 (655 Å$^2$ for K1 and 556 Å$^2$ for K2), suggesting that the interaction of the toxin with the surrounding membrane is important for stabilizing the toxin-channel complex.

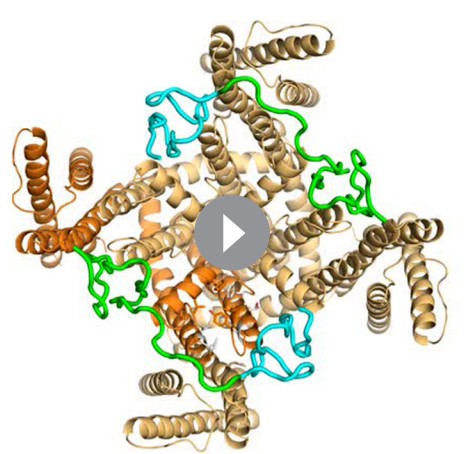

**Video 3.** Structural changes in TRPV1 resulting from DkTx binding, and close-up view of a cluster of buried hydrophobic residues that become exposed upon channel opening.

It is interesting to compare the present structural picture of DkTx binding to TRPV1 with the mode of PcTx1 binding to ASIC, as well as that described for tarantula toxins binding to voltage-sensing domains in Kv channels. DkTx, PcTx1 and voltage-sensor toxins like hanatoxin and GxTx-1E adopt similar folds, and display amphipathic surfaces that enable interactions with lipid membranes (*Figure 8*) (*Gupta et al., 2015*). In DkTx, residues in loops 2 and 4 engage the outer pore of TRPV1, while interacting with lipids in the surrounding membrane (*Figures 3*, *4*, *8A,B*), and contain mostly hydrophobic residues (compare *Figure 8A,B* with *Figure 1C,D* where the toxins are oriented identically). In PcTx1, residues in loops 1 and 4 form a clamp-like structure for the toxin to bind to helix-5 within the extracellular thumb domain of the ASIC channel (*Figure 8D*) (*Baconguis and Gouaux, 2012*; *Dawson et al., 2012*). Interestingly, the voltage-sensor toxin GxTx-1E also

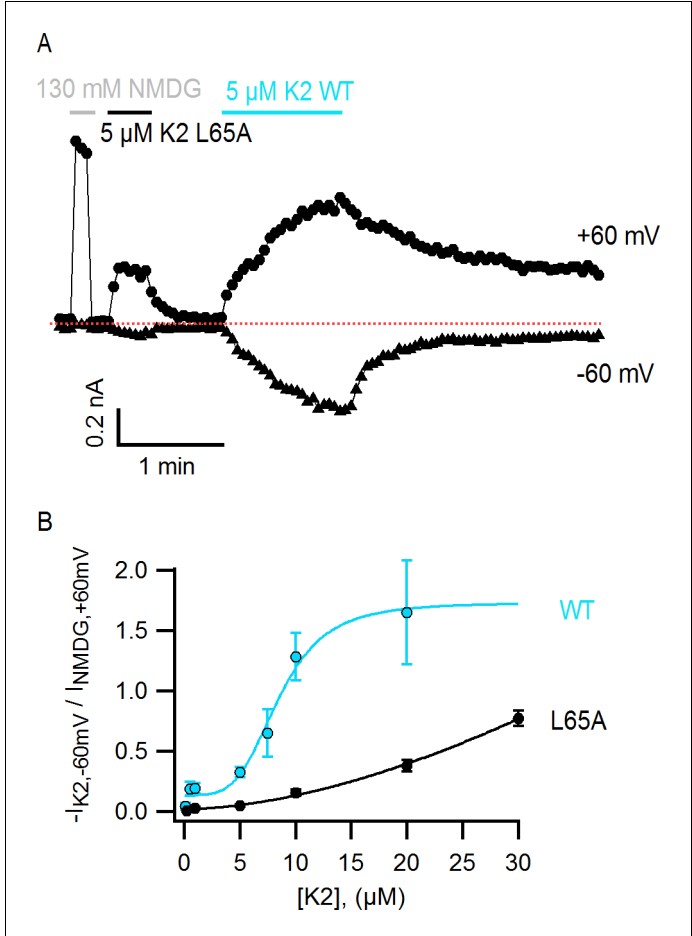

**Figure 7.** The L65A mutation in K2 perturbs activation of the TRPV1 channel. (**A**) Representative TRPV1 current time-course in HEK293 cells depicting activation of the channel by L65A and WT K2 toxins in the whole-cell configuration at +60 and −60 mV. Currents were measured every 3 s by applying voltage-ramps of 1 s duration from −120 to +140 mV starting at a holding potential of −90 mV. Currents were recorded under constant perfusion with control solution or with the test solutions containing either NMDG$^+$ instead of Na$^+$ as permeant ion (gray), or the K2 toxins together with 130 mM NaCl (black and blue), as indicated by the continuous horizontal lines. The dotted red line denotes the zero-current level. Removing external Na$^+$ leads to constitutive activation of TRPV1 to a similar extent as a saturating concentration of DkTx or K2 (*Jara-Oseguera, 2016*), and therefore provides a convenient means of normalizing response to the toxins. (**B**) Normalized concentration-response relation for TRPV1 activation at −60 mV by WT K2 toxin (blue) or the L65A mutant (black) obtained in the whole-cell configuration using HEK293 cells. Currents at −60 mV were measured from voltage-ramps as in (**A**), and normalized to the current value at +60 mV measured before application of the toxin, in the presence of an external solution containing 130 mM NMDGCl instead of NaCl (see (**A**), gray). Initial currents measured in control solution at +60 and −60 mV were subtracted from the data. A single toxin variant (WT or L65A) was evaluated per experiment to construct dose-response curves, and one or more toxin concentrations were tested per cell. Data are shown as mean ± sem, with n = 3–14 for each data point. The continuous curves are fits to the Hill equation with parameters: WT, $K_D$ = 8.5 μM, n = 4.0; L65A, $K_D$ > 0.2 mM, n = 1.7.

employs residues in loops 1 and 4 to bind to the S3b helix within the voltage-sensing domain of Kv channels, and these form a surface that resembles the one used by PcTx1 to bind to ASIC (*Figure 8E*) (*Gupta et al., 2015*). If we orient the two lobes of DkTx with channel-binding surfaces directed down, as if viewing the toxin-channel complex from a side view in the membrane, and oriented both PcTx1 and GxTx-1E by backbone superposition (as is done in *Figure 8A,B,D,E*), the channel-binding surfaces for PcTx1 and GxTx-1E would both be displaced laterally by roughly 90° (compare the location of residues highlighted in red in *Figure 8A,B* for K1 and K2 with *Figure 8D*

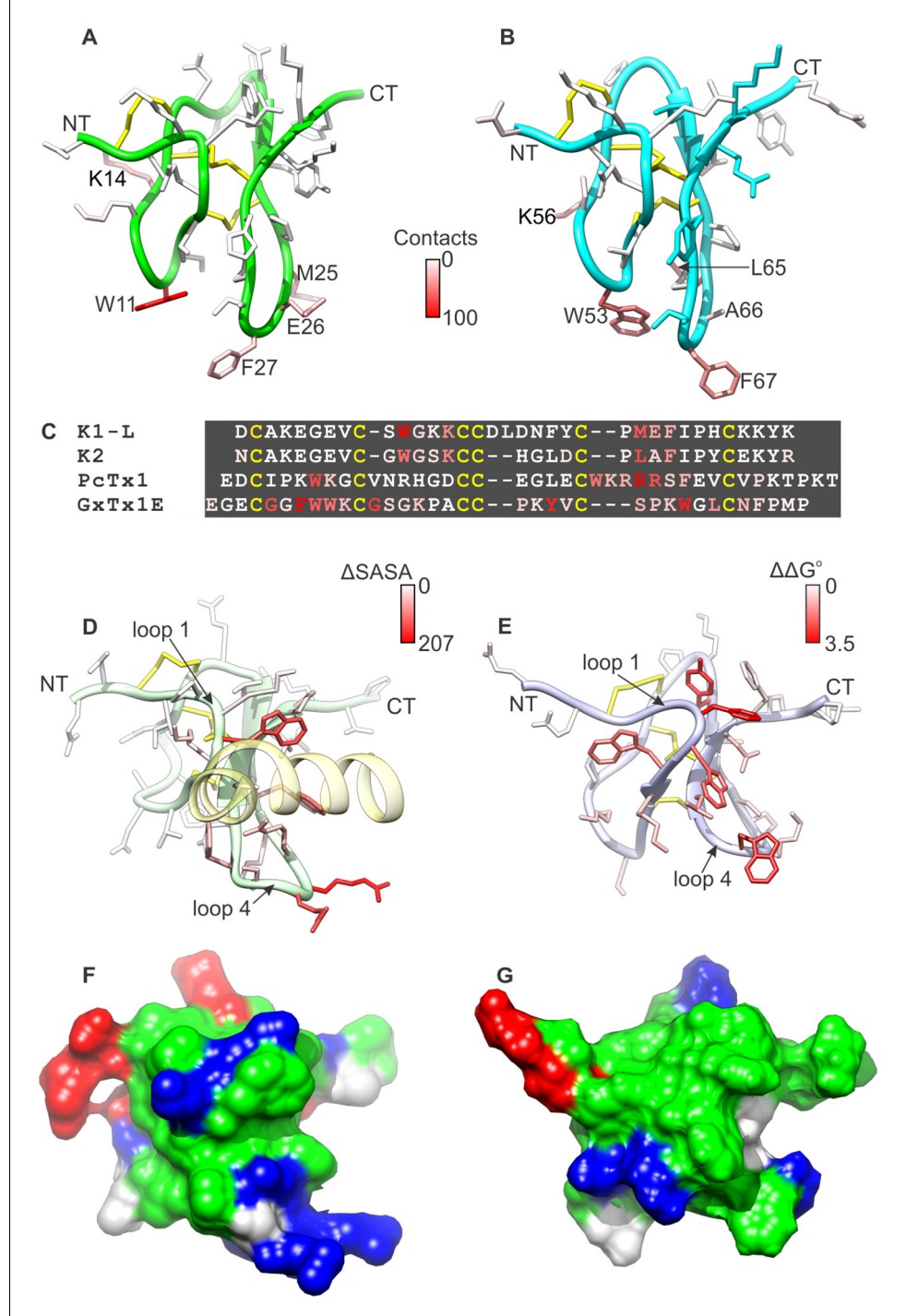

**Figure 8.** Comparison of active surfaces of different classes of tarantula toxins. (**A,B**) Active surfaces of K1 (**A**) and K2 (**B**). Side-chains in DkTx are colored based on the probability of finding TRPV1 channel residues within 6 Å in a simulation of the complex (except cysteine residues colored in yellow). The value specified in the scale bar reflects the number residues of the channel with one or more atoms within 6 Å of the toxin residue, on average. Orientation of K1 and K2 are the same as in **Figure 1A–D**, with loops 2 and 4 pointing down. (**C**) Sequences of tarantula toxins. Conserved cysteine residues are shown in yellow and residues are colored according to schemes in **A**, **B**, **D** or **E** as appropriate. (**D**) Structure of PcTx1 bound to helix-5 of the ASIC1a channel (pastel yellow) oriented the same as K1 and K2 in **A** and **B** by superimposing backbones (PDB entry 4FZ0). Backbone of the toxin is colored in pastel green, and side-chains are colored by changes in solvent-accessible surface area between the structure of PcTx1 when the toxin is bound to ASIC1a and the structure of toxin without channel (**Gupta et al.,**

*Figure 8 continued on next page*

*Figure 8 continued*

**2015**) (ΔSASA in Å$^2$). (**E**) Solution structure of GxTx1E oriented the same as K1 and K2 in **A** and **B** by superimposing backbones (PDB entry 2WH9). Side-chains are colored according to perturbation in free energy of binding (ΔΔG° in kcal mol$^{-1}$) (*Gupta et al., 2015*). (**F**) Surface representation of the PcTx1 structure where hydrophobic residues are colored green, basic residues blue and acidic residues red. Orientation is the same as in **D**. (**G**) Surface representation of the GxTx-1E structure where hydrophobic residues are colored green, basic residues blue and acidic residues red. Orientation is the same as in **E**.

for PcTx1 and *Figure 8E* for GxTx-1E). Interestingly, the amphipathic surface of GxTx-1E is similarly displaced relative to the amphipathic surfaces of both lobes of DkTx, which makes sense because those surfaces for both toxins are important for interacting with membranes and their target channels. Overall, this comparison suggests that the common fold of all these toxins provides at least two distinct surfaces for engaging with ion channel proteins, while maintaining favorable interactions with the lipid membrane.

Our results also provide important insight into the bivalent structure of DkTx. Previous studies have demonstrated that the K2 lobe is a more effective activator of TRPV1 when compared to K1 (see also *Figure 3F*), and that tethering of the two lobes of DkTx dramatically enhances the lifetime of the toxin-channel complex (*Bae et al., 2012*; *Bohlen et al., 2010*), similar to the avidity effect of a bivalent antibody. We find that the two lobes of DkTx also have very different energetics for partitioning into lipid membranes, with the K1 lobe displaying considerably stronger interactions when compared to K2, a trend that can be recapitulated in the bivalent K1K1 construct when compared to the bivalent K2K2 construct (*Figure 5*). Using bivalent K1K1 and K2K2 constructs, we also found that differences in binding affinity can explain why K2 is a much better activator of TRPV1 compared to K1 (*Figure 3D*). Thus, bivalency not only increases the lifetime of the toxin-channel complex, but also enables the two lobes to be separately tuned for optimal membrane partitioning or affinity for binding to the channel. This feature of DkTx would suggest that the toxin preferentially uses the K1 lobe to interact with membranes and thus raise the local concentration of K2 at the membrane surface, thereby promoting the initial formation of the toxin-channel complex using K2 within the interfacial region of the membrane (*Figure 9*). It will be fascinating to further explore the structural and biophysical basis of these mechanisms of recognition, to better understand how protein-protein interactions are modulated by the membrane environment.

In docking of DkTx into the cryo-EM maps of TRPV1 and optimizing the structures of both the toxin-bound and apo channels, we identified an interesting hydrophobic cluster behind the SF of TRPV1 that undergoes a noticeable conformational change. This cluster appears to be a determining factor in the energetics of channel opening, as several mutations in this cluster diminish toxin activation of the channel even though most do not directly contact the toxin (*Bohlen et al., 2010*). It is tantalizing to speculate that the disruption of this cluster, which implies an increased solvent exposure of more than a dozen hydrophobic side-chains across four channel subunits, leads not only to the structural changes required for ion permeation, but also to the kind of increase in the heat capacity of the protein that has been postulated to explain temperature sensing in TRP channels, including TRPV1 (*Clapham and Miller, 2011*). Notably, mutations at positions T633, F640 and Y653 within the hydrophobic cluster have been shown to alter temperature sensitivity of the TRPV1 channel without affecting activation of the channel by capsaicin (*Grandl et al., 2010*; *Myers et al., 2008*; *Ryu et al., 2007*). In addition, it is interesting that an external Na$^+$ binding site in the outer pore of TRPV1 has been shown to tightly regulate temperature-sensor activation, and that DkTx renders TRPV1 temperature-insensitive over a wide temperature range (*Jara-Oseguera, 2016*). The hypothesis that this rearranging hydrophobic cluster is involved in temperature sensing will require careful experimental examination, and could begin to provide a unifying framework to rationalize how very different stimuli cooperate to regulate TRP channel activation.

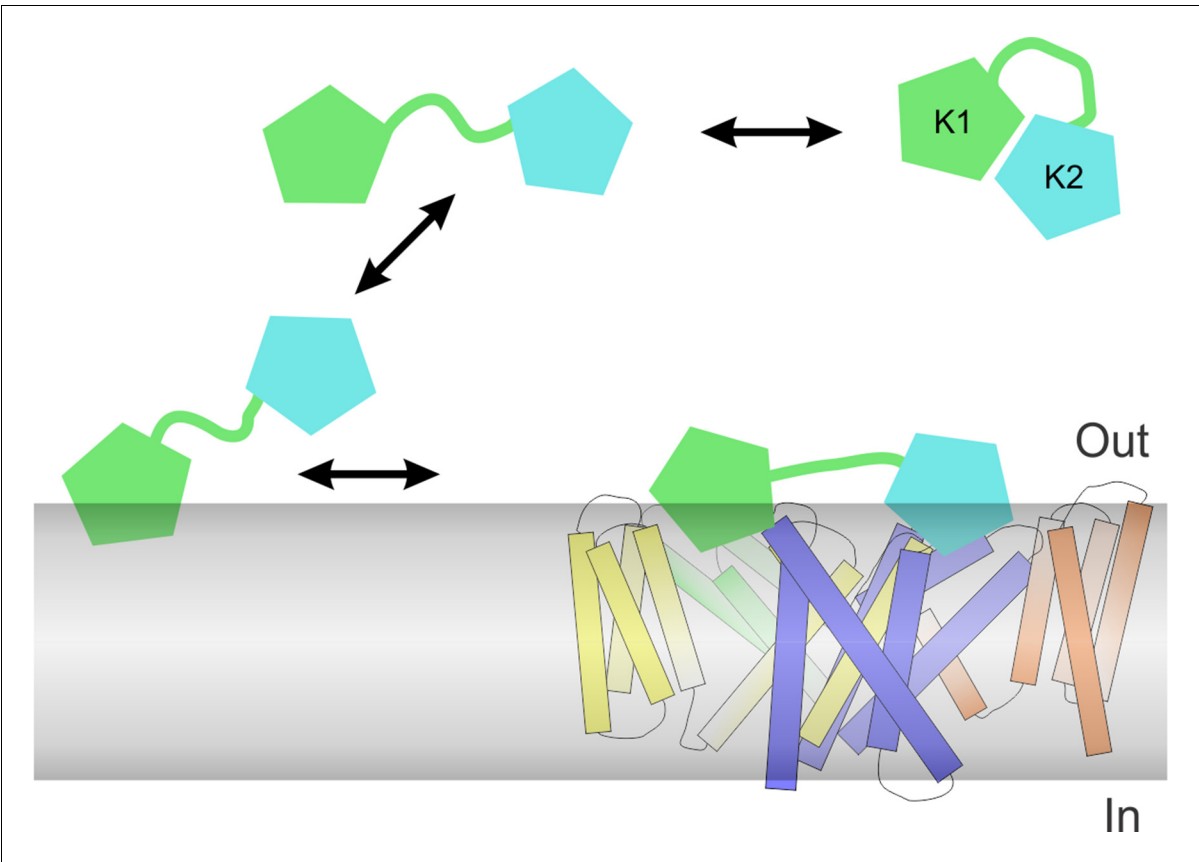

**Figure 9.** Proposed mechanism of DkTx partitioning into membranes and binding to TRPV1. Partitioning of DkTx into the lipid membrane (gray) before recognition of TRPV1 is preferentially driven by the K1 lobe (green), whereas K2 (cyan) has a higher affinity for binding to the channel. After binding to TRPV1, both K1 and K2 retain a significant number of interactions with the lipid membrane, presumably so as to stabilize the complex, which features a relatively small protein-protein interface.

# Materials and methods

## Sample preparation and NMR spectroscopy

K1, K2 and DkTx were prepared as previously described (*Bae et al., 2012*). The sequence of bivalent K1 (K1K1) is: GDCAKEGEVCSWGKKCCDLDNFYCPMEFIPHCKKYKPYVPVTTDCAKEGEVCSWGKKCC-DLDNFYCPMEFIPHCKKYK. The toxin was prepared using a protocol similar to DkTx (*Bae et al., 2012*), where K1K1 were produced in *E. coli* as a fusion protein with ketosteroid isomerase (KSI). K1K1 was cleaved from KSI with hydroxylamine, refolded and purified using reversed-phase HPLC. The sequence of bivalent K2 (K2K2) is: NCAKEGEVCGWGSKCCHGLDCPLAFIPYCEKYRPYVPVTTNC-AKEGEVCGWGSKCCHGLDCPLAFIPYCEK. The toxin was prepared by cloning synthetic K2K2 DNA into the pET28a vector with an additional Met added at the N-terminus of the K2K2 sequence. The toxin was produced in *E. coli* as a 6 Histidine tagged form, refolded and purified using Ni-affinity chromatography. The N-terminal fusion protein was removed by CNBr cleavage and the toxin purified using reversed-phase HPLC. For experiments shown in *Figure 5C*, K2 was prepared using an identical protocol to that for K2K2, but an additional Met was inserted before the N-terminus of N41 in the K2K2 construct such that CNBr cleavage would yield K2. $^{15}$N labeled DkTx was prepared by producing the toxin in M9 minimal media where $^{15}$NH$_4$Cl was included as a nitrogen source, and otherwise prepared as previously described (*Bae et al., 2012*).

K1 and K2 were dissolved at concentrations of 1 mM in 90% H$_2$O/10% D$_2$O (pH 4.0) containing trimethylsilyl propionate (TSP) as an internal standard, and double-quantum-filtered correlation spectroscopy (DQF-COSY), total correlation spectroscopy (TOCSY) and nuclear Overhauser effect spectroscopy (NOESY) spectra recorded at 298 K or 310 K in a Bruker 600 MHz spectrometer. Mixing

times of TOCSY and NOESY experiments were 80 ms and 250 ms, respectively. $^{15}$N labeled DkTx was dissolved at a concentration of 0.4 mM in 10 mM sodium phosphate buffer (pH 4) containing 10% $D_2O$ and TSP as an internal standard, and $^1$H-$^{15}$N NOESY-heteronuclear single quantum coherence (HSQC) and $^1$H-$^{15}$N TOCSY-HSQC spectra recorded at 298 K in a 900 MHz Bruker DRX900 spectrometer equipped with cryogenic probe. All spectra were processed with NMRPipe (*Delaglio et al., 1995*), and analyzed and assigned with NMRview (*Kirby et al., 2004*).

## NMR structure calculation

J-coupling constants were estimated from DQF-COSY spectra (*Kim and Prestegard, 1989*) and imposed as dihedral angle restraints for structure calculation based on the following rules: for $^3J_{HNH\alpha}$ values of <5.5 Hz, the phi angle was constrained in the range of $-65 \pm 25°$; For $^3J_{HNH\alpha}$ values of >8.0 Hz, the phi angle was constrained in the range of $-120 \pm 40°$. Interproton distance restraints were obtained from unambiguous NOE peaks that were manually assigned using NMRview (*Kirby et al., 2004*). The initial structure of the toxins were generated by a simulated annealing protocol in torsion angle space using Cyana2.1 (*Guntert et al., 1997*) by imposing interproton distance and dihedral angle restraints. The structure was further refined using Xplor-NIH 2.37 (*Bermejo et al., 2012*; *Schwieters et al., 2006*), using interproton distances, dihedral angle restraints, and disulfide bond restraints based on sequence homology with other ICK toxins were imposed along with a multidimensional torsion angle potential of mean force (*Bermejo et al., 2012*), a gyration volume term to enforce proper packing (*Schwieters and Clore, 2008*) and standard covalent and nonbonded energy terms. The quality of each of the 20 ensemble structures were analyzed by Protein Structure Validation Software suite (http://psvs-1_5-dev.nesg.org). Analysis of Ramachandran plots with Procheck gave the following results: for K1, most favored region (76.8%), additionally allowed regions (22.2%), generously allowed regions (0.8%) and disallowed regions (0.2%); for K2, favored region (58.2%), additionally allowed regions (31.4%), generously allowed regions (6.4%) and disallowed regions (4.1%).

## Initial docking of K1 and K2 using Xplor-NIH

Xplor-NIH was used to dock the NMR structures of K1 and K2 into a cryo-EM map for TRPV1 bound to DkTx/RTx with imposed twofold symmetry (kindly provided Dr. Yifan Cheng and colleagues). Energy-minimized structures of K1 and K2 were placed in the EM map in a clockwise (CW) or counter-clockwise (CCW) configuration, and combined with a truncated model of the TRPV1 structure comprising residues 518–550 and 585–645 from one chain and 644–669 from the adjacent chain (from PDB entry 3J5Q). Electron density map fitting was achieved using the probDistPot energy term (*Gong et al., 2015*) with a force constant of 100 kcal/mol throughout. Energy-minimized structures of the truncated complex were determined by imposing electron density map along with interproton distance restraints of K1 and K2 from solution NMR experiments. In addition to these experimental terms, the multidimensional torsion angle potential of mean force, empirical backbone hydrogen bonding potential of mean force (*Grishaev and Bax, 2004*) and standard covalent and nonbonded terms were employed. 20 structures were calculated and the 10 lowest energy models were then used for further analysis.

## Refinement of TRPV1 and TRPV1-DkTx structures using ROSETTA

To prepare the structures of apo TRPV1 and the TRPV1-DxTx complex for refinement, we first modeled the missing loops, side-chains and termini into the existing structures (PDB entries 3J5P and 3J5Q), using MODELLER version 9.10 (*Eswar et al., 2006*); for the TRPV1-DxTx complex, the truncated Xplor-NIH models (CCW and CW) were merged with the published channel structure. The fit-to-density protocol (*DiMaio et al., 2009*) of ROSETTA version 2014wk05 (*Leaver-Fay et al., 2011*) was then used to refine the three models (apo and DxTx-bound, CW or CCW). In a first stage, 100 models were generated for each system following the 'relax' application in 5 cycles, using a high-resolution energy function for proteins in the membrane environment (*Barth et al., 2007*; *Yarov-Yarovoy et al., 2006*) in addition to the experimental cryo-EM maps. The fit-to-density score was determined with a 9-residue sliding-window, and was added to the total ROSETTA score with a weight of 0.2. The symmetry of the cryo-EM maps was explicitly imposed on all the models; a four-fold symmetric map was used for apo TRPV1, and a twofold symmetric map for the DkTx-TRPV1

complex. Each of these 100 models required approximately 280 computer-hours on an Intel Xeon 2.4 GHz core-processor for the apo TRPV1 channel and 800 hr for the complex. The models with the best ROSETTA score for apo TRPV1 and TRPV1 with DxTx in a CCW configuration were then used as the seed for a second refinement stage. In this stage, positional restraints were imposed on the backbone and on well-resolved side-chains, and the configuration of all remaining side-chains was refined through extensive sampling and scoring. In particular, we ranked the degree of confidence in the position of each side-chain atom by evaluating a score equal to $-m_a\rho(x_a, y_a, z_a)$, where $m_a$ is the mass of the atom and $\rho(x_a, y_a, z_a)$ the normalized density signal at the hypothetical atom position (*Wu et al., 2013*). Atoms for which the calculated score was in the top 20% were considered to be well resolved and were not modified further. To generate a plausible configuration for the remaining side-chain atoms, we generated 12,000 new models, using the ROSETTA energy function only (i.e. without a fit-to-map restraint). On average, each new model of apo TRPV1 and TRPV1-DkTk required approximately 15 and 25 core-minutes, respectively. To identify the most representative model for each of these ensembles, we used the clustering algorithm of Daura and colleagues as implemented in GROMACS version 4.6.5 (*Daura et al., 1999*; *Hess et al., 2008*), using a similarity cut-off of 1 Å for apo TRPV1 and 0.08 Å for the toxin-channel complex. The two selected models correspond to the central structure of the most populated cluster obtained in each case (~1,100 and ~7,900 structures, respectively).

## Molecular dynamics simulations of TRPV1 and TRPV1-DkTx in a membrane

The improved structures of apo and DkTx-bound TRPV1 were embedded in a cubic simulation box of side length of 145 Å containing a bilayer of 1-palmitoyl-2-oleoyl-sn-glycero-3-phosphocholine (POPC) lipids, surrounded by water. The simulation systems were prepared using GRIFFIN (*Staritzbichler et al., 2011*). The set of protonation states in the channel and toxin was selected on the basis of a Monte-Carlo/Poisson-equation simulation, as described previously (*Eicher et al., 2014*), for neutral pH. Energetically favorable positions for buried water molecules within the protein were identified with DOWSER (*Zhang and Hermans, 1996*). $Na^+$ and $Cl^-$ ions were introduced to achieve electroneutrality and a 100 mM electrolyte concentration. Each of the simulation systems amounts to ~315,000 atoms in total, including over 500 lipid molecules and ~69,000 water molecules.

The simulations were carried out using the CHARMM36 force-field for proteins and lipids (*Best et al., 2012*; *Klauda et al., 2010*) as implemented in NAMD version 2.9 (*Phillips et al., 2005a*), at constant pressure and temperature, and with periodic boundary conditions. The equations of motion were integrated with a time-step of 2 fs. The pressure (1 atm) was maintained constant with a Nose-Hoover Langevin-piston barostat (*Feller et al., 1995*), allowing variations in the volume of the simulation cell but keeping a constant ratio in the membrane-plane dimensions. The temperature (298 K) was maintained with a Langevin thermostat. Electrostatic interactions were calculated using Particle-Mesh Ewald (PME) (*Darden et al., 1993*) with a real space cutoff of 12 Å. The same cut-off was used for truncating van-der-Waals interactions, modeled with a shifted Lennard-Jones potential. To equilibrate the simulation systems, we used conventional MD simulations with gradually weaker positional restraints applied to the protein, the toxin and buried water molecules over 12 ns. For the toxin-channel complex, a subsequent MD simulation was carried out for 400 ns using the twofold symmetric cryo-EM map of the complex as a three-dimensional restraint, via the MDFF module in NAMD (*Trabuco et al., 2009*) (with 0.3 kcal/mol as the scaling factor). During the last 200 ns of this simulation, we coupled the $\chi_1$ and $\chi_2$ torsions of all side-chains at the toxin-channel interface to a fictitious temperature of 3,000 K, using an extended-Lagrangian approach (*Iannuzzi et al., 2003*), so as to accelerate the configurational sampling of that interface. The set of accelerated side-chains was determined by analyzing the first 200 ns of simulation using a simple definition of toxin-channel contacts based on a distance cut-off of 3.5 Å; this set comprises all the residues specified in *Figure 4*. Finally, snapshots from the last 200 ns of simulation were analyzed to identify the most persistent contacts between toxin, channel and membrane. A similar scheme was followed for the apo channel, comprising an MDFF simulation of 200 ns (using the corresponding fourfold symmetric cryo-EM map), with enhanced side-chain sampling in the last 100 ns, which were considered for analysis.

To quantify the significance of a hypothetical contact between a given side-chain in the channel or toxin (group of atoms A) and one or more side-chains in the other protein, or the head-group and/or tail regions of the lipid membrane (group of atoms B), we calculated the non-normalized distribution function of the distance $r$ between each of the atoms in group A and each of the atoms in group B (only non-hydrogen atoms were considered). This collective distribution function $g(r)$ was then integrated over the first shell of interaction, which we assumed to be 6 Å:

$$N_p = \int_0^6 4\pi r^2 g(r) \, dr$$

The resulting value, denoted by $N_p$, therefore represents the number of contacting atom pairs in groups A and B; note this value scales with the size of the groups, but we reasoned that so does the strength of their interaction. All analyses were performed with VMD version 1.9.1 (*Humphrey et al., 1996*).

## Computational alanine-scanning of the toxin-channel interface

The computational alanine-scanning of the residues at the toxin-channel interface was carried out with ROSETTA version 2014wk05 (*Kortemme et al., 2004*; *Leaver-Fay et al., 2011*). Specifically, the algorithm was used to independently identify the toxin residues at the interface with the channel, to replace these residues individually with alanine, and to estimate the effect of this mutation on the binding free energy of the complex. Positive values of this estimate imply that the alanine mutation is predicted to destabilize the complex, while negative values imply a stabilizing effect. The scan was carried for 200,000 different input configurations, extracted from the last 200 ns of the MD simulation of the complex; the results for each toxin residue were then averaged. After each alanine substitution, interfacial residues within 10 Å were re-optimized using the ROSETTA repacking method.

## Membrane interaction of toxin

Large unilamellar vesicles (LUVs) were prepared by extruding lipid suspensions made of a 1:1 mix of 1-palmitoyl-2-oleoyl-sn-glycero-3-phosphocholine (POPC) and 1-palmitoyl-2-oleoyl-sn-glycero-3-[phospho-rac-(1-glycerol)] (POPG) through 100 nm polycarbonate film. Toxins (2 μM or 5 μM in 10 mM HEPES, 1 mM EDTA, pH 7.0 buffer) in cuvette were excited with 280 nm wavelength and emission spectra were measured from 300 to 450 nm in the presence or absence of LUV using a SPEC FluoroMax. Scattering of light from lipid vesicles were corrected (*Gupta et al., 2015*; *Ladokhin et al., 2000*; *Milescu et al., 2007*). Smooth curves were obtained by fitting the following partition function, $F/F_0(L) = 1+(F/F_0^{max}-1)K_x[L]/([W]+K_x[L])$, to the data points, where $F/F_0(L)$ is relative fluorescence intensity of 320 nm at a given lipid concentration, $F/F_0^{max}$ is fluorescence intensity when partitioning is saturated, [L] is molar concentration of accessible lipid (60% of total lipid, corresponding to the outer leaflet), [W] is molar concentration of water (55.3 M), and $K_x$ is mole-fraction partition coefficient.

## Two-electrode voltage clamp recordings

*Xenopus laevis* oocytes were surgically removed and gently shaken for 60 min in a solution of 82.5 mM NaCl, 2.5 mM KCl, 1 mM MgCl$_2$, 5 mM HEPES and 2 mg/mL collagenase. A rat TRPV1 construct (generously provided by D. Julius, UCSF) was cloned into the pGEM-HE vector, and used to generate cRNA. The cRNA was then injected into oocytes, which were then incubated for 1–3 days at 17°C in ND-96 solution (96 mM NaCl, 2 mM KCl, 1.8 mM CaCl$_2$, 5 mM HEPES, 1 mM MgCl$_2$ and 50 μg/mL gentamycin, titrated to pH 7.6 with NaOH). TRPV1 activity was recorded under voltage clamp using a two-electrode voltage clamp (OC-725C; Warner Instruments) in a 150-μL recording chamber. The recorded data were filtered at 1 kHz and digitized at 5 kHz using a digidata analog/digital converter and pClamp software (Molecular Devices). Microelectrode resistances were 0.1–1 MΩ when filled with 3 M KCl. The external recording solution contained 115 mM NaCl, 2.5 mM KCl, 1.5 mM MgCl$_2$ and 10 mM HEPES, titrated to pH 7.4 with NaOH. All experiments were performed at room temperature (~22°C).

## Whole-cell patch clamp recordings

HEK293 cells were transiently transfected with rTRPV1 and Green Fluorescence Protein (pGreen-Lantern, Invitrogen) cDNAs using FuGENE6 (Roche) transfection reagent following manufacturer's instructions, and used for recording 12–24 hr after transfection. Standard whole-cell patch clamp recordings at room temperature (22–24°C) were performed. Data was acquired with an Axopatch 200B amplifier (Axon Instruments), filtered with an 8-pole low-pass Bessel filter (model 900, Frequency Devices) and digitized with a Digidata 1322A interphase and pClamp10 software (Axon Instruments). All data was analyzed using Igor Pro 6.34A (Wavemetrics Inc.). Pipettes were pulled from borosilicate glass and heat-polished to final resistances between 2–4 MΩ. 80–95% series resistance (Rs) compensation was used. An agar bridge (1 M KCl, 3% agarose) was used to connect the recording chamber with the ground electrode. A holding potential of –90 mV was used in all experiments. Data were acquired at 10 kHz and low-pass filtered at 2 kHz. For the voltage-ramps, voltage was stepped down from the holding to −120mV for 50 ms, then ramped up to +140 mV in 1s and returned to −90 mV for 50 ms. A ramp was applied every 3 s. Recordings were done using isometric solutions consisting of (in mM): 130 NaCl, 10 HEPES, 10 EGTA, pH 7.4. Solutions were applied using a gravity-fed rapid solution exchange system (RSC-200, BioLogic). Cells were lifted from the coverslip and placed in front of glass capillaries perfused with the different solutions. Fused silica tubing (250 µM internal diameter, Polymicro Technologies) was used to deliver toxin. Fresh toxin solutions were prepared every day. Dose-response curves were fit to the Hill equation:

$$\frac{I_{Tx}}{I_{NMDG}} = I_{min} + \frac{I_{max} - I_{min}}{1 + \left(\frac{K_D}{[Tx]}\right)^s},$$

where $I_{Tx}$ is the current measured at the various toxin concentrations at −60 mV, $I_{NMDG}$ is the current measured in the presence of 130 mM NMDGCl at +60 mV, $I_{min}$ is the minimal current remaining after subtraction of the currents in control conditions (i.e. 130 mM NaCl before application of the toxins), $I_{max}$ is the maximal activation achieved by the toxins relative to 130 mM NMDGCl, $K_D$ is the apparent dissociation constant, $[Tx]$ is the molar concentration of the toxin and $s$ is the Hill coefficient.

## Acknowledgements

We thank members of the Swartz lab for helpful discussions. This work was supported by the Intramural Research Programs of the NINDS, NHLBI, and CIT, NIH to KJS, JDG, and CDS, respectively; by the Basic Science Research Program through the National Research Foundation of Korea, funded by the Ministry of Education, Science and Technology (2013R1A1A2009798); by the Bio Imaging Research Center at GIST to JIK; and by a grant from the KRIBB Research Initiative Program (Korean Biomedical Scientist Fellowship Program) of the Korea Research Institute of Bioscience and Biotechnology to C.B. The computational work in this study was carried out in part using the Biowulf high-performance computing cluster hosted at the National Institutes of Health, Bethesda, MD.

## Additional information

### Competing interests

KJS: Reviewing editor, *eLife.* The other authors declare that no competing interests exist.

### Funding

| Funder | Author |
| --- | --- |
| National Institute of Neurological Disorders and Stroke | Kenton J Swartz |
| National Heart, Lung, and Blood Institute | José D Faraldo-Gómez |
| Center for Information Technology | Charles D Schwieters |

| National Research Foundation of Korea | Jae Il Kim |
|---|---|
| Korea Research Institute of Bioscience and Biotechnology | Chanhyung Bae |

The funders had no role in study design, data collection and interpretation, or the decision to submit the work for publication.

## Author contributions

CB, Conception and design, Acquisition of data, Analysis and interpretation of data, Drafting or revising the article, Contributed unpublished essential data or reagents; CA, AJO, Conception and design, Acquisition of data, Analysis and interpretation of data, Drafting or revising the article; JK, DK, CWL, E-HK, Conception and design, Acquisition of data, Analysis and interpretation of data; CDS, JDFG, Conception and design, Analysis and interpretation of data, Drafting or revising the article; JIK, KJS, Conception and design, Analysis and interpretation of data, Drafting or revising the article, Contributed unpublished essential data or reagents

## Author ORCIDs

Claudio Anselmi, http://orcid.org/0000-0002-3017-5085
José D Faraldo-Gómez, http://orcid.org/0000-0001-7224-7676
Kenton J Swartz, http://orcid.org/0000-0003-3419-0765

## Ethics

Animal experimentation: This study was performed in strict accordance with the recommendations in the Guide for the Care and Use of Laboratory Animals of the National Institutes of Health. All of the animals were handled according to approved institutional animal care and use committee (IACUC) protocol (#1253-15) of the National Institute of Neurological Disorders and Stroke.

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
