## [Decision Letter]

Thank you for submitting your work entitled "Structure of a double-knot tarantula toxin bound to the TRPV1 channel at the protein-lipid interface" for consideration by *eLife*. Your article has been reviewed by three peer reviewers, and the evaluation has been overseen by Richard Aldrich as the Senior Editor. Two of the three reviewers have agreed to reveal their identity: Vincenzo Carnevale and Katie Henzler-Wildman (peer reviewers).

The reviewers have discussed the reviews with one another and the Reviewing editor has drafted this decision to help you prepare a revised submission.

Summary:

The manuscript by Bae et al. reports a structural characterization of the complex between the TRPV1 ion channel and the double-knot tarantula toxin. Building on the recent work by Yifan Cheng and colleagues, the authors adopted an array of approaches (both experimental and computational) to attain a description of the channel-toxin interactions with an atomic level of detail.

The authors use solution NMR to determine the structure of the individual lobes of Double Knot Toxin DkTx, then use ROSETTA to model DkTx into the cryo-EM map of the TRPV1/DkTx complex. Examination of the structure obtained in this way reveals significant lipid-toxin interaction and a relatively small toxin-channel interface as compared to other tarantula toxins. The authors confirm the lipid partitioning of the toxin by monitoring intrinsic Trp fluorescence, finding that K1 partitions more strongly into the membrane. In contrast, K2 has a greater affinity for TRPV1, a slower off-rate as monitored in electrophysiology experiments. The authors take advantage of this difference in affinity to confirm the specific toxin loops that interact with the channel using loop-swapping experiments in single-knot constructs. This leads to a novel mechanism, where one knot is optimized for lipid partitioning while the other knot is optimized for channel interaction.

The overall picture is insightful and compelling and rests on solid experimental observation and computational results. The resulting molecular model advances our structural knowledge of TRPV1 and enables other investigators to further study the properties of this channel. Most importantly, this study constitutes a much-needed stepping-stone to address the relevant question of the molecular mechanism of TRPV1 activation. The research has been rigorously conducted and described with a level of detail that is appropriate to ensure reproducibility of the results, yet still comprehensible by non-specialists. The discussion of the results is thoughtprovoking and of potential interest to a large community of scientists.

Essential revisions:

The paper could and should be made stronger by reexamining the data and providing improved discussions, without additional lengthy experiments. The required revisions fall into four general areas:

1) There should be a more extensive comparison of apo and toxin-bound channel structure with emphasis on the potential mechanism of activation.

2) The discussion on the comparison with other toxins should be removed or it should be clarified, as it is currently confusing. This needs a clearer explanation and perhaps a larger reference frame or cartoon model in Figure 6 to provide a clearer perspective.

3) More extensive discussion on the role of lipid partitioning of toxin constructs. Partition experiments could be conducted on chimeric mutants analyzed in Figure 3. The partition data on these mutants may provide stronger evidence on the existing correlation (or anticorrelation) between lipid partitioning and affinity for the toxin. Such experiments should, in principle, be straightforward.

4) The question of whether toxin lobes interact intra-molecularly in two-lobe toxin constructs or even inter-molecularly in single-lobe constructs should be addressed. This comment is driven by the expected energies of interactions (based on Figure 5) suggesting that closed structures (or dimers) should predominate. This would require additional experiments (unless there are inter domain NOEs in the data that has already been collected), although RDC data on the 15N samples already prepared for the HSQC spectra shown in the current manuscript should be relatively straightforward to perform.

---

## [Author Response]

Essential revisions:The paper could and should be made stronger by reexamining the data and providing improved discussions, without additional lengthy experiments. The required revisions fall into four general areas:1) There should be a more extensive comparison of apo and toxin-bound channel structure with emphasis on the potential mechanism of activation.

We very much appreciate the opportunity to put forward our mechanistic interpretation of the newly refined structural data. At this point, our proposal is necessarily hypothetical, but nevertheless we feel that it will be of interest to the readers, and that it will help design new experimental work.

Specifically, we have added a new section to the revised version of the manuscript in which we highlight the major differences between these two structures, as well as the mechanism of channel activation that can be inferred from those differences. Given the limited resolution of the structures, we have also evaluated the statistical significance of our key observations using molecular dynamics simulations. MD simulations of apo TRPV1, analogous to those previously presented for the toxin-bound channel, are now also included in our study. A new figure has been added to the revised manuscript to summarize our observations (see Figure 6)

This comparative analysis indicates that binding of DkTx promotes changes in the transmembrane architecture of the channel that pertain not only to the internal structure of each of the channel subunits, but also to their relative arrangement; as a result, these seemingly highly collective structural changes open up the constrictions observed in the apo state, in the selectivity filter (SF) and the intracellular gate.

These structural changes appear to be effected through displacements in the pore helix (P) and S6 helices, relative to the S1-S4 unit and the transmembrane segment of S5, both of which are largely unchanged. That the P and S6 helices become displaced, relative to S1-S5, is consistent with the fact that the N-termini of P and S6, respectively, are the primary contacts for the toxin on the channel extracellular surface. The changes in P and S6 within a given subunit correlate very clearly with a pronounced rearrangement of the SF and pore loop in that same subunit, but also with changes in the intracellular side of the channel. The rationale for this remote effect is that the P and S6 helices are the main interfacial elements between adjacent subunits, in the transmembrane region of the channel; therefore, displacements in these elements are propagated to the adjacent subunits, adding to the changes in their own internal structure. As a result, the S1-S4 units (which as mentioned remain largely unchanged) become noticeably displaced relative to each other, thus affecting the crossing angle of helices S5 and S6, and thus the degree of opening of the intracellular gate.

Very interestingly, the displacements induced by DkTx on the channel main-chain correlate with the disruption of a cluster of hydrophobic interactions behind the SF, at the interface between S5, the pore helix and S6, in close proximity to the contact region with the toxin. This network involves, among others, residues I599, F659 and V595, F649, as well as T650. Only the latter two residues contacts directly DkTx in the toxin-bound structure; however, mutation of any of these residues diminishes DkTx-induced opening of the channel, particularly in the case of I599A, F649A and F659A. Therefore, it seems clear that the stability of this hydrophobic cluster is a determining factor for channel activation. It is tantalizing to speculate that the disruption of this cluster, which implies an increased solvent exposure of more than a dozen hydrophobic side-chains across the four channel subunits, leads not only to the structural changes required to allow for ion permeation, but also to the kind of increase in the heat capacity of the protein that has been postulated to underlie temperature sensing in TRP channels – i.e. the changes just described might explain, at least in part, the temperature dependence of the activation process of TRPV1.

The relevant changes to the manuscript can be found in the subheading “Membrane partitioning of DkTx” of the Results, paragraphs one to three of the Discussion, in new Figure 6 and Video 3, and are reflected in the new title and Abstract. We also include new data on a toxin mutant (Figure 7) to provide experimental support for some residues in the hydrophobic cluster interacting with specific toxin residues.

2) The discussion on the comparison with other toxins should be removed or it should be clarified, as it is currently confusing. This needs a clearer explanation and perhaps a larger reference frame or cartoon model in Figure 6 to provide a clearer perspective.

We have revised the text (Discussion), modified the figure (now Figure 8) and its legend to make it clear that all toxins are shown with the same orientation (by backbone superposition) and that channel binding surfaces are on present on different parts of the toxins.

3) More extensive discussion on the role of lipid partitioning of toxin constructs. Partition experiments could be conducted on chimeric mutants analyzed in Figure 3. The partition data on these mutants may provide stronger evidence on the existing correlation (or anticorrelation) between lipid partitioning and affinity for the toxin. Such experiments should, in principle, be straightforward.

We did not originally undertake these experiments because we exhausted our supply of the K1(K2L2) chimera performing functional studies. We synthesized a new batch of this chimera and completed the requested partitioning experiments. Unfortunately they are not very informative. Although these chimeras were originally designed to identify regions in K2 that would enhance the affinity of K1 (gain of function), from the perspective of membrane partitioning they can only be used to identify regions that weaken partitioning (loss of function). The new results show that transfer of the N-terminal residue, or loops 2, 3 or 4, either leave K1’s strong partitioning intact or paradoxically enhance the strength of partitioning. Transfer of the C-terminus greatly reduces the blue shift observed with partitioning, either because it weakens partitioning or because it alters the orientation of the toxin (and the relevant Trp residue) within the membrane. Unfortunately we cannot readily distinguish between these possibilities because the small blue-shift would make it impossible to construct a titration. We include these data in a new supplementary figure (Figure 5—figure supplement 1), and comment briefly on them in the Results section in paragraph one of subheading “Membrane partitioning of DkTx”. Since none of the chimeras weakened the strong partitioning of K1, they don’t provide the basis to establish a correlation or anti-correlation between partitioning and affinity, nor do they define regions of the toxins most important for membrane partitioning.

4) The question of whether toxin lobes interact intra-molecularly in two-lobe toxin constructs or even inter-molecularly in single-lobe constructs should be addressed. This comment is driven by the expected energies of interactions (based on Figure 5) suggesting that closed structures (or dimers) should predominate. This would require additional experiments (unless there are inter domain NOEs in the data that has already been collected), although RDC data on the 15N samples already prepared for the HSQC spectra shown in the current manuscript should be relatively straightforward to perform.

Although the question of whether the lobes interact in the context of the membrane merits further examination, we consider it to be tangential to the current study. This possibility was discussed because we noticed a systematic energy penalty for bivalency in the toxin-membrane interactions, and we felt compelled to comment. We have revised this section to be clear about why we think there may be inter-lobe interactions, despite the fact that we cannot see NOEs in the DkTx spectra. To our thinking, if DkTx containing the natural linker and the version with a 7-Gly linker have the same penalty, it would seem that there must inter-lobe interactions. We no longer have 15N samples to undertake RDC experiments, and we would need to prepare fresh samples, and at least for us, the requested experiments would require considerable optimization.